# STFlow: Data-Coupled Flow Matching for Geometric Trajectory Simulation

Kiet Bennema ten Brinke [1]   Koen Minartz [1]   Vlado Menkovski [1]

## Abstract

Simulating trajectories of dynamical systems is a fundamental problem in a wide range of fields such as molecular dynamics, biochemistry, and pedestrian dynamics. Machine learning has become an invaluable tool for scaling physics-based simulators and developing models directly from experimental data. In particular, recent advances in deep generative modeling and geometric deep learning enable probabilistic simulation by learning complex trajectory distributions while respecting intrinsic permutation and time-shift symmetries. However, trajectories of N-body systems are commonly characterized by high sensitivity to perturbations leading to bifurcations, as well as multi-scale temporal and spatial correlations. To address these challenges, we introduce STFlow *(Spatio-Temporal Flow)*, a generative model based on graph neural networks and hierarchical convolutions. By incorporating data-dependent couplings within the Flow Matching framework, STFlow denoises starting from conditioned random-walks instead of Gaussian noise. This novel informed prior simplifies the learning task by reducing transport cost, increasing training and inference efficiency. We validate our approach on N-body systems, molecular dynamics, and human trajectory forecasting. Across these benchmarks, STFlow achieves the lowest prediction errors with fewer simulation steps and improved scalability. Code is available at github.com/MrKietho/STFlow 

## 1. Introduction

N-body systems arise in a wide range of scientific and societal domains, including astrophysics (Heggie, 2003), molec-

[1]Machine Learning for Physical Sciences (ML4Sci/e) Group, Department of Mathematics & Computer Science, Eindhoven University of Technology, The Netherlands. Correspondence to: Kiet Bennema ten Brinke <k.bennema.ten.brinke@tue.nl>.

*Proceedings of the 43rd International Conference on Machine Learning*, Seoul, South Korea. PMLR 306, 2026. Copyright 2026 by the author(s).

ular dynamics (Thompson et al., 2022) and human mobility (Minartz et al., 2025). Faithfully simulating their evolution enables the study of complex collective phenomena and support decision making in settings where physical modeling is infeasible or prohibitively expensive. However, these systems are characterized by strong interactions, multi-scale temporal dependencies and frequent sensitivity to perturbations, making trajectory prediction both computationally and statistically challenging. As a result, feasible simulators must scale to large systems while modeling uncertainty over future dynamics.

Recent work has therefore increasingly framed trajectory simulation as a conditional generative modeling problem. Autoregressive models provide a flexible framework but suffer from error accumulation and limited parallelism (van den Oord et al., 2016; Hoogeboom et al., 2022). One-shot generative approaches, most prominently diffusion and flow-matching models, alleviate these issues and have demonstrated strong performance on high-dimensional temporal data (Han et al., 2024). Nevertheless, most existing methods rely on uninformed Gaussian priors in their forward processes.

This choice induces a large transport cost between the prior and target trajectory distribution, forcing the learning dynamics to perform substantial denoising. In practice, this leads to complex vector fields, high numbers of function evaluations during simulation, and sensitivity to hyperparameter tuning (Liu et al., 2023; Lipman et al., 2023). Importantly, in conditional trajectory simulation rich information about the system's future evolution is already available through observed initial frames. Yet, current approaches largely fail to exploit this information when constructing the prior distribution.

In this work, we argue that incorporating conditioning information directly into the generative prior is essential for efficient and scalable conditional trajectory simulation. We introduce Spatio-Temporal Flow (STFlow), a probabilistic simulation model for fixed-length geometric trajectories based on Flow Matching (Liu et al., 2023; Lipman et al., 2023). STFlow lowers simulation errors and improves training and inference efficiency by employing data-dependent couplings. This coupling is used to construct informed random-walk priors that simplify the mapping between the

prior and data distributions by reflecting the statistics of the dynamics present in the conditioning signal. STFlow combines this strategy with a permutation- and time-shift equivariant spatiotemporal architecture. Our main contributions are summarized as follows:

- We introduce STFlow, a data-driven simulator that learns a permutation-invariant probability distribution over the dynamics of a set of particles. STFlow utilizes convolutional and message passing neural network components to effectively model spatio-temporal dependencies while scaling linearly in both the number of particles and trajectory length.

- We propose a simple yet effective prior distribution for the conditional trajectory simulation setting based on data-dependent couplings. By constructing the prior as a random walk with parameters estimated from the trajectory's observed timesteps, the resulting samples are naturally coupled during training and inference. This allows for effective learning of a straighter flow from prior to data, requiring only a few integration steps during inference whilst increasing simulation fidelity.

- We evaluate our approach on three challenging datasets: a synthetic physics-based N-body system dataset, human trajectory forecasting, and molecular dynamics simulation. Our results show that STFlow produces the lowest prediction errors on these benchmarks in almost all experiments. Moreover, we validate scalability, we display goodness-of-fit and we investigate the importance of the coupled prior and architectural design choices, and find that these are key drivers of the observed performance improvements.

## 2. Related Work

Early work in the domain of deep learning-driven geometric trajectory simulation generally used models based on autoregressive Graph Neural Networks (GNNs), as they align with the permutation-symmetric nature of the problem (Battaglia et al., 2016; Kipf et al., 2018; Sanchez-Gonzalez et al., 2020). Since then, various architectural and modeling advances have been proposed, for example focusing on continuous-time dynamics through Neural ODEs (Gupta et al., 2022; Luo et al., 2023; 2024), the incorporation of symmetry constraints (Satorras et al., 2021; Thomas et al., 2018b; Brandstetter et al., 2022a; Fuchs et al., 2020; Ruhe et al., 2023; Bekkers et al., 2024), or on interpretability through symbolic methods (Cranmer et al., 2020; Lemos et al., 2023).

Although the above methods are relevant to geometric trajectory simulation, many real-life systems are stochastic

or otherwise do not lend themselves to point prediction methods, for example due to their chaotic nature. Consequently, recent works have proposed probabilistic simulation methods based on generative models. We distinguish two main categories of approaches. First, autoregressive probabilistic models generate trajectories through a recurrent time-stepping approach (Flunkert et al., 2017; Salzmann et al., 2020; Gupta et al., 2018; Amirian et al., 2019; Xu et al., 2022b; Yıldız et al., 2022; Minartz et al., 2023; Wu et al., 2023). Although this method is in principle capable of real-time generation of indefinitely long trajectories, it suffers from error accumulation as the simulation horizon increases (Sanchez-Gonzalez et al., 2020; Brandstetter et al., 2022b; Minartz et al., 2023; Lippe et al., 2023) and is difficult or impossible to parallelize during both training and inference, limiting their scalability.

Recent trajectory generative models that synthesize whole trajectories in one shot (e.g. diffusion-based methods) achieve high precision but typically rely on unconditioned Gaussian forward processes, increasing transport cost and requiring large numbers of denoising steps (Han et al., 2024; Jiang et al., 2023). Flow matching variants reduce this cost but predominantly still uses Gaussian priors or does not exploit conditioning information to construct the prior (Liu et al., 2023; Albergo et al., 2024). Finally, several recent models set aside efficiency by not enforcing permutation/time-shift symmetries (Sestak et al., 2025; Fu et al., 2025) or use architectures whose time and memory scale poorly in trajectory length (Han et al., 2024). In contrast, STFlow explicitly uses a prior that conserves observed frames, reduces transport cost, provides a flexible design space for conditioning and pairs this with an efficient scalable spatio-temporal architecture, see the ablations (§4.4) and scaling experiments (§4.3) that quantify these benefits.

## 3. STFlow: generative modeling of geometric trajectories using data-dependent couplings

### 3.1. Problem formulation

**Geometric trajectories.** This work focuses on trajectories embedded in Euclidean space, specifically geometric trajectories, which we define as $\mathcal{G} := (\mathbf{x}, \mathbf{h}, \mathcal{E})$, where $\mathbf{x} := \mathbf{x}^{0:T} := \{[\mathbf{x}_{(i)}^0, ..., \mathbf{x}_{(i)}^{T-1}]\}_{i=0}^N \in \mathbb{R}^{T \times N \times d}$ is a set of $N$ objects containing the sequence of their coordinates in $d$ dimensional space over $T$ timesteps, $\mathbf{h} := \{[\mathbf{h}_{(i)}^0, ..., \mathbf{h}_{(i)}^{T-1}]\}_{i=0}^N \in \mathbb{R}^{T \times N \times d_h}$ are node features with dimensionality $d_h$ that serve as model input, and $\mathcal{E}$ is the set of edges defining the interaction structure between the objects within the same timestep. $c$ is the number of initial frames that are observed and serve as input to the simulation model, and $f := T - c$ is the number of target frames. Parameters $c$ and $f$ are fixed per dataset.

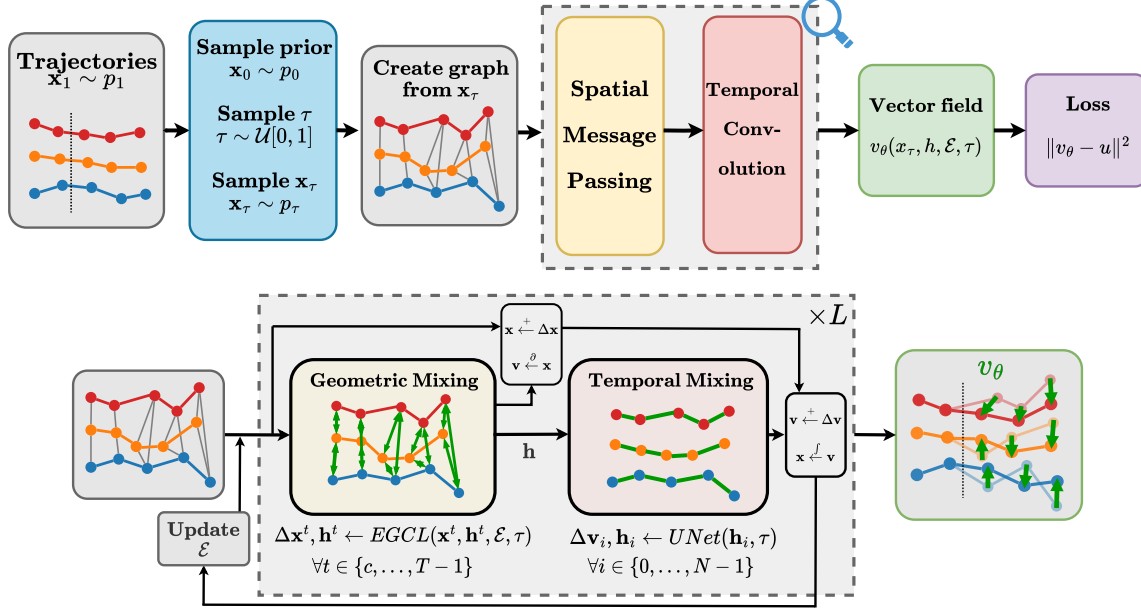

*Figure 1.* Overview of STFlow. Given trajectories $\mathbf{x}_1 \sim p_1$, we construct a noisy prior $\mathbf{x}_0$ informed by the observed initial conditions and predict a vector field $v_\theta$ using repeating layers of spatial message passing and temporal convolution.

**Goal and approach.** Our goal is to train a data-driven simulation model by learning the joint probability distribution over trajectories $\mathbf{x}$ from the ground truth distribution $p_1$ given $\mathbf{h}$ and $\mathcal{E}$:

$$p_1(\mathbf{x} \mid \mathbf{h}, \mathcal{E}) = p_1(\mathbf{x}^{c:T} \mid \mathbf{x}^{0:c}, \mathbf{h}, \mathcal{E}) \, p_1(\mathbf{x}^{0:c} \mid \mathbf{h}^{0:c}, \mathcal{E}^{0:c}) \tag{1}$$

In the simulation problem setting, we presuppose that an initial part of the trajectory $\mathbf{x}^{0:c}$ is provided to the model as initial conditions, such that we can readily obtain samples from the ground truth $p_1(\mathbf{x}^{0:c} | \mathbf{h}^{0:c}, \mathcal{E}^{0:c})$. Consequently, the machine learning task boils down to learning the distribution over future timesteps $p_1(\mathbf{x}^{c:T} | \mathbf{x}^{0:c}, \mathbf{h}, \mathcal{E})$.

We limit ourselves to conditioning in a forward-temporal formulation. However, we note that our approach can perform the task of data assimilation in wide array of settings, where conditioning information can be present at any timestep. This includes no observed frames specifying unconditional generation, $c = T - 1$ for an autoregressive formulation or the task of inpainting or super-resolution where observations and targets are interleaved, among others.

We approach this problem by leveraging flow matching (Liu et al., 2023; Lipman et al., 2023). Flow matching involves training a model $v_\theta$ to approximate a ground truth time-dependent *vector field* $u : \mathbb{R}^{T \times N \times d} \times [0,1] \to \mathbb{R}^{T \times N \times d}$. This vector field defines a *flow* $\psi$ from a known prior distribution $p_0$ to the target data distribution $p_1$. The flow is distributed according to a *probability path* $(p_\tau)_{0 \leq \tau \leq 1}$, from which we can sample trajectories $\mathbf{x}_\tau$ that flow from the prior to the target distribution given $\tau$ from 0 to 1. Inference is

carried out by sampling noisy trajectories $\mathbf{x}_0 \sim p_0$ from the prior and then solving the Ordinary Differential Equation (ODE) $d\mathbf{x} = v_\theta(\mathbf{x}, \mathbf{h}, \mathcal{E}, \tau)d\tau$ determined by the learned vector field $v_\theta$.

### 3.2. Flow Matching with data-dependent coupling

**Data-dependent couplings** In order to train the model we have to draw samples $(\mathbf{x}_0, \mathbf{x}_1) \sim \pi$ from some coupling $\pi$ of the source distribution $p_0$ and target distribution $p_1$. The simplest option would be to draw these pairs independently, i.e. $\pi = p_0(\mathbf{x}_0)p_1(\mathbf{x}_1)$, with $p_0$ being a simple distribution such as a Gaussian, as is the standard approach in diffusion-based models. However, this approach would ignore the fact that there is a natural coupling present our problem setting. Instead, we define $p_0$ through a *data-dependent coupling* $\pi(\mathbf{x}_0, \mathbf{x}_1) = \pi(\mathbf{x}_0|\mathbf{x}_1)p_1(\mathbf{x}_1)$ (Albergo et al., 2024). Such a data-dependent coupling defines a prior density $\int_{\mathbf{x}_1} \pi(\mathbf{x}_0|\mathbf{x}_1)p_1(\mathbf{x}_1)d\mathbf{x}_1$ over $\mathbf{x}_0$. Let $\pi(\mathbf{x}_0|\mathbf{x}_1)$ be of the form $\mathbf{x}_0 = m(\mathbf{x}_1) + \zeta$, where $m : \mathbb{R}^{T \times N \times d} \to \mathbb{R}^{T \times N \times d}$ is a function that corrupts $\mathbf{x}_1$ and $\zeta$ is a stochastic variable that is conditionally independent of $\mathbf{x}_1$ given $m(\mathbf{x}_1)$. Following the factorization of Equation 1, we choose $m(\mathbf{x}_1) = [\mathbf{x}_1^0, ..., \mathbf{x}_1^{c-1}, \mathbf{0}, ..., \mathbf{0}]$ and $\zeta = [\mathbf{0}, ..., \mathbf{0}, \zeta^c, ..., \zeta^{T-1}]$. Intuitively, this represents a prior in which the $c$ initial observations remain intact in $\mathbf{x}_0$, while the to-be-simulated part of the trajectory is replaced with stochastic variables $\zeta^t$.

**Informed prior** A straightforward choice for a distribution over $\zeta$, and also the one used in Albergo et al. (2024), would be a multivariate Gaussian $\mathcal{N}(0, \Sigma)$. However, the flow $\psi$

from $p_0(\mathbf{x}_0)$ to $p_1(\mathbf{x}_1)$ is easier to learn when the *transport cost* is minimized (Tong et al., 2024; Albergo et al., 2024). To achieve this, we require a prior $p_0$ which **1.** incorporates trajectory-specific characteristics to reduce transport cost via initializations that are closer to $\mathbf{x}_1$ in space; **2.** is computationally efficient to sample from; and **3.** is conditionally independent of $\mathbf{x}_1^{c:T}$ given $m(\mathbf{x}_1)$. With this in mind, and with $\mathbf{x}_0 := [\mathbf{x}_1^0, ..., \mathbf{x}_1^{c-1}, \zeta^c, ..., \zeta^{T-1}]$ we set $\zeta \in \mathbb{R}^{f \times N \times d}$ to be distributed as follows:

$$\zeta^t = \zeta^{t-1} + \mu + \sigma_o \odot \mathbf{z}^t \qquad \forall t \in \{c, ..., T-1\},$$
$$\zeta^{c-1} = \mathbf{x}^{c-1}$$

$$(2)$$

where $\mu = \frac{1}{c-1}\sum_{t=1}^{c-1}(\mathbf{x}^t - \mathbf{x}^{t-1})$, $\sigma_o = \frac{1}{c-1}\sum_{t=1}^{c-1}\|\mathbf{x}^t - \mathbf{x}^{t-1} - \mu\|^2 \cdot s$ with $\mu, \sigma_0 \in \mathbb{R}^{N \times d}$, where $s$ is a hyperparameter controlling the additional velocity variance in $p_0$, $\odot$ denotes element-wise multiplication, and $\mathbf{z}^t \sim \mathcal{N}(\mathbf{0}, \mathbf{I})$. Now, $\zeta$ as defined in Equation 2 is a random walk of which the parameters are fitted to the initial observed part of each node's trajectory. As such, this random walk obeys basic physical properties of trajectories, such as continuity and inertia. Our hypothesis holds, that by constructing the prior to be more aligned with realistic dynamics, the learning task is simplified and the probability paths $p_\tau$ are shorter, which expedites both the training efficiency and simulation accuracy.

**Training** The conditional flow matching training objective consists of *matching* the *flows* by training the vector field $v_\theta$ using Mean Squared Error (MSE) (Liu et al., 2023; Lipman et al., 2023). We conventionally assume linear Gaussian probability paths, such that $\mathbf{x}_\tau \sim p_\tau := \mathcal{N}(\mathbf{x} \mid \tau\mathbf{x}_1 + (1-\tau)\mathbf{x}_0, \sigma_p^2)$. This lets the vector field $u$ to take the simple form of $u = \mathbf{x}_1 - \mathbf{x}_0$ (Tong et al., 2024). In our case, the loss $\mathcal{L}_{CFM}$ is defined as:

$$\mathcal{L}_{CFM}(\theta) = \mathbb{E}_{\tau, \mathbf{x}_0, \mathbf{x}_1}\|v_\theta(\mathbf{x}_\tau, \mathbf{h}, \mathcal{E}, \tau) - (\mathbf{x}_1 - \mathbf{x}_0)\|^2 \quad (3)$$

where $\tau \sim \mathcal{U}[0,1]^\alpha$, $\mathbf{x}_1 \sim p_1$, $\mathbf{x}_0 = m(\mathbf{x}_1) + \zeta$, and $\mathbf{x}_\tau \sim p_\tau$. Conventionally, $\alpha = 1$ for a uniform $\tau$ distribution, we noticed however that for $\alpha = 0.5$, skewing the distribution towards $\tau = 1$, tends to increase performance. During training, we provide our model with the mask indicating which frames are considered observed, which is used for generating $\mathbf{x}_0$. The output of $v_\theta$ at timesteps $t \in \{0, .., c-1\}$ is ignored, as only the parts of the trajectories which are not already observed are predicted. STFlow also allows for training with varying observed window size $c$, such that trajectories can be sampled both unconditionally or with any amount of observed timesteps, without additional retraining or guidance needed. A detailed description of the training procedure is denoted in A.1.

**Inference** We sample noisy trajectories from the prior $\mathbf{x}_0 \sim p_0$ and integrate the learned vector field $v_\theta(\mathbf{x}_\tau, \mathbf{h}, \mathcal{E}, \tau)$ using the Euler integration method with only 5 function eval-

uations (NFEs) to simulate trajectories $\mathbf{x}^{c:T}$ given $\mathbf{x}^{0:c}$ as initial conditions. This is presented in A.1 for more detail.

### 3.3. Model Architecture

The two main building blocks of the architecture for $v_\theta(\mathbf{x}, \mathbf{h}, \mathcal{E}, \tau)$ are the *Spatial message passing* (SMP) layer and the *Temporal mixing* (TM) layer, visualized in Figure 1. Our SMP layer is based on the Equivariant graph convolutional layer (EGCL) by Satorras et al. (2021) as used in Han et al. (2024), which captures spatial interactions by aggregating learned messages along the edges. The layer is defined as $\Delta\mathbf{x}^t, \mathbf{h}^t \leftarrow \text{EGCL}(\mathbf{x}^t, \mathbf{h}^t, \mathcal{E}^t, \tau)$, learning the change in positions $\Delta\mathbf{x}^{c:T}$ and node features in a permutation-equivariant fashion. We set the graph topology to be function of node distances, which together with the edge features are recalculated after every SMP layer, making use of the new denoised positions. Message passing between nodes is conducted for each timestep independently. The EGCL layer fuses the geometric information contained in the graph, informing nodes of their spatial neighborhood.

The Temporal mixing layer consists of the UNet from Dhariwal & Nichol (2021) designed for diffusion on images with built-in conditioning on noise level $\tau$. Defined as $\Delta\mathbf{v}^{c:T}, \mathbf{h} \leftarrow \text{UNet}(\mathbf{h}, \tau)$, it predicts the required change in velocities across all timesteps, in a residual-learning style. The function of the UNet is to capture dynamics of individual trajectories and to propagate the conditioning signal over learned representations $\mathbf{h}$. It uses weight sharing along the time dimension introducing time-shift equivariance, it learns multi-scale representations and has inductive bias towards local patterns. These properties align with the locally correlated, multi-scale and time-homogeneous dynamics we are interested in. Spatial and Temporal layers are alternated to create the architecture of STFlow parameterizing $v_\theta$, more details are in Appendix A.2. In addition we experiment with using a modern Transformer (Labs et al., 2025) as Temporal layer instead to assess how global self-attention compares to more local convolutions in this setting. An overview of properties of STFlow compared with relevant state-of-the-art approaches is shown in Table 1.

**Computational Complexity** The Spatial message passing layer has computational complexity of $\mathcal{O}(|\mathcal{E}|d^2)$ (Gilmer et al., 2017). The Temporal UNet layer scales linearly in number of frames, namely $\mathcal{O}(NTd^2)$, as we make use of 1D convolutions on each trajectory. Consequently, STFlow has a forward-pass computational complexity of $\mathcal{O}(L(|\mathcal{E}|d^2 + NTd^2)) = \mathcal{O}(LNTd^2)$ for number of layers $L$, since the graph connectivity is based on a fixed distance threshold.

*Table 1.* Property overview of relevant state-of-the-art approaches.

| | GeoTDM[1] | MoFlow[2] | LaM-SLidE[3] | STFlow |
|---|---|---|---|---|
| Permutation equivariant | ✓ | ✗ | ∼ | ✓ |
| Time-shift equivariant | ✓ | ✗ | ✗ | ✓ |
| Data-dependent coupling | ✗ | ✗ | ✗ | ✓ |
| Scalable in trajectory length | ✗ | - | ∼ | ✓ |
| Fast inference (low NFEs) | ✗ | ✓ | ∼ | ✓ |

[1] Han et al. (2024), [2] Fu et al. (2025), [3] Sestak et al. (2025)

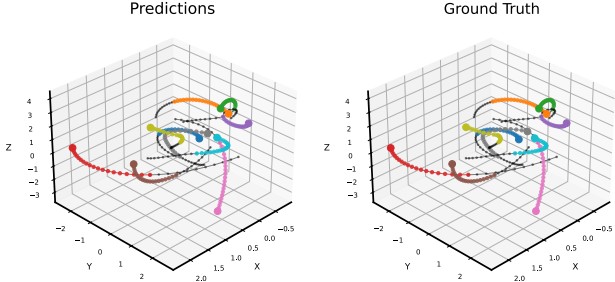

*Figure 2.* Inference results on the N-Body gravity dataset. The black dots represent the 10 conditioning steps, the colored dots are the 20 generated steps.

## 4. Experiments

The experimental evaluation of STFlow is performed on N-body physical simulation (§4.1.1), molecular dynamics (§4.1.2) and human trajectory forecasting (§4.1.3) to show generalization capability across multiple domains of the conditional trajectory prediction setting. In each domain we measure displacement errors of the generated trajectories and compare them with other relevant approaches. We also evaluate on velocity and acceleration densities of the simulated trajectories (§4.2) to assess goodness-of-fit, we present how the model scales in time and memory with respect to the system's size (§4.3) and ablate on core components of the approach (§4.4). The used hyperparameters are denoted in A.3, compute resources in A.4 and additional density evaluations and trajectory visualizations in A.6.

### 4.1. Conditional trajectory generation

#### 4.1.1. N-BODY SYSTEMS

**Datasets and setup** As part of the first benchmark, three different kinds of simulated N-body systems are used, each with different forces acting on $N$ particles. **1.** *Charged Particles* (Kipf et al., 2018; Satorras et al., 2021) where $N = 5$ particles with random charges of $-1$ or $+1$ are driven by the Coulomb force **2.** *Spring Dynamics* (Kipf et al., 2018) containing $N = 5$ particles with random mass which are connected by springs with a probability of 0.5 between pairs, where the springs abide by Hooke's Law. **3.** *Gravity System* (Brandstetter et al., 2022a) with $N = 10$ particles, each having a random mass and initial velocity influenced by gravitational forces. All trajectories have a total length of $T = 30$ with a conditioning window of $c = 10$. For all three datasets the split consists of 3000 training, 2000 validation and 2000 testing samples.

**Baselines** We look at three different kinds of approaches. Frame-to-frame point-prediction models: Tensor field networks (Thomas et al., 2018a), SE(3)-Transformers (Fuchs et al., 2020) and EGNN (Satorras et al., 2021); Deterministic trajectory models: Eqmotion (Xu et al., 2023) and Proba-

*Table 2.* Conditional generation on N-Body systems, results are averaged over 5 runs.

| | Charged | | Spring | | Gravity | |
|---|---|---|---|---|---|---|
| | ADE | FDE | ADE | FDE | ADE | FDE |
| Prior $p_0$ | 0.526 | 1.013 | 0.2312 | 0.5083 | 1.591 | 3.101 |
| TFN[*] (2018b) | 0.330 | 0.754 | 0.1013 | 0.2364 | 0.327 | 0.761 |
| SE(3)-Tr[*] (2020) | 0.395 | 0.936 | 0.0865 | 0.2043 | 0.338 | 0.830 |
| EGNN[*] (2021) | 0.186 | 0.426 | 0.0101 | 0.0231 | 0.310 | 0.709 |
| EqMotion[*] (2023) | 0.141 | 0.310 | 0.0134 | 0.0358 | 0.302 | 0.671 |
| GeoTDM[*] (2024) | 0.110 | 0.258 | 0.0030 | 0.0079 | 0.256 | 0.613 |
| LaM-SLidE (2025) | 0.104 | 0.238 | 0.0070 | 0.0135 | 0.157 | 0.406 |
| STFlow | 0.085 | 0.158 | **0.0004** | **0.0008** | 0.107 | 0.231 |
| STFlow-Transformer | **0.079** | **0.151** | 0.0007 | 0.0013 | **0.049** | **0.109** |

[*] results reported from Han et al. (2024)

bilistic trajectory models: GeoTDM (Han et al., 2024) and LaM-SLidE (Sestak et al., 2025). To put the problem complexity into perspective, we also report the metrics when using samples from our prior $p_0$ (Eq. 2) as predictions.

**Metrics** As often used in trajectory forecasting literature, we look at the Average Displacement Error (ADE) and Final Displacement Error (FDE) to measure performance. ADE is defined as $ADE(\mathbf{x}^{c:T}, \mathbf{y}^{c:T}) = \frac{1}{N(T-c)} \sum_{t=c}^{T-1} \sum_{i=0}^{N-1} \|\mathbf{x}_i^t - \mathbf{y}_i^t\|_2$ and FDE as $FDE(\mathbf{x}^{c:T}, \mathbf{y}^{c:T}) = \frac{1}{N} \sum_{i=0}^{N-1} \|\mathbf{x}_i^{T-1} - \mathbf{y}_i^{T-1}\|_2$. For probabilistic models, the mean ADE and FDE of 5 runs for each test sample is reported, aligning with other works.

**Implementation** The particles are represented as fully connected dynamic graphs $\mathcal{G}$, where each particle at each timestep has a node. Edges only exist between nodes in the same timestep. Node features consist of position, velocity and velocity magnitude, as well as acceleration and acceleration magnitude; the edge features contain Euclidean distance, relative position and relative velocity. In the Spring setting, a value of 0 or 1 is added as edge feature indicating a spring and in the Charged setting the charge is added as node feature, which are also used in the baselines. We pre-

dict a vector field of velocities instead of positions, use 3 layers of STFlow with a hidden dimension of 64, use only 5 number of function evaluations during inference and use data augmentation during training by random rotations.

**Results** The results are shown in Table 2 and a visualization of the gravity dataset can be seen in Figure 2. STFlow produces lower errors compared to all other models, where the biggest difference can be seen in the FDE metric. The performance increase compared to the previous best evaluated model is on average $48.1\%$ for ADE and $56.9\%$ for FDE, while using $2\times$ less NFEs than the previously best performing model LaM-SLidE (Sestak et al., 2025) and $200\times$ less than GeoTDM (Han et al., 2024). Using a Transformer as Temporal layer significantly improves results in the Gravity setting, but errors are higher in the Springs case.

### 4.1.2. MD17

**Datasets and setup** For the simulation of molecule trajectories we evaluate on the MD17 dataset (Chmiela et al., 2017). The dataset contains 8 small molecules from which positions over time have been simulated in one long trajectory using Density Functional Theory (Kohn et al., 1996). Each molecule has 9 (Ethanol, Malonaldehyde) to 21 (Aspirin) atoms and each molecule has a different trajectory length. We subsample by keeping 1 out of every 10 frames as in Han et al. (2024) and use a training/validation/test split of $70/15/15$, split along the time axis. Trajectory lengths are $T = 30$ with $c = 10$ and a step size of 10 frames is used to sample the trajectories. The same baselines and metrics as the N-body experiments are used here.

**Implementation** We again use a fully connected graph, as the number of nodes is small, instead of using $n$-hop connectivity. We predict a vector field on the atom velocities. As node features we add a one hot encoding representing the atom type. We do not use any additional edge features. We use 3 layers of STFlow with a hidden dimension of 64, use 5 NFE during inference and use data augmentation during training by random rotation along the three axis.

**Results** The results are plotted in Table 3. STFlow produces the lowest errors across 7 of 8 molecules, with an average reduction of $15.6\%$ ADE and $22.6\%$ FDE compared to LaM-SLidE (Sestak et al., 2025), the previous best scoring model for all molecules. Using a transformer as Temporal layer gives $16.3\%$ ADE and $18.5\%$ FDE decrease. Both results demonstrate that our model is capable of learning complex molecular dynamics without domain-specific features.

*Table 4.* Conditional generation on the NBA dataset for two setups. **1.** Two scenario's with metrics $\min_{20}$ADE / $\min_{20}$FDE in feet, first 8 frames (0.96s) are conditioning and 12 frames (1.44s) are predicted. **2.** Both $mean_{20}$ and $\min_{20}$ ADE/FDE as metrics, measured in meters. Here, 10 frames (2.0s) are given and 20 frames (4.0s) are predicted.

| **1** | Rebound | Score |
|---|---|---|
| Trajectron++ [*](Salzmann et al., 2020) | 0.98/1.93 | 0.73/1.46 |
| SVAE [*](Xu et al., 2022b) | **0.72/1.37** | **0.64**/1.17 |
| LaM-SLidE (Sestak et al., 2025) | 0.79/1.42 | **0.64/1.09** |
| STFlow | 0.94/1.55 | 0.82/1.35 |
| STFlow-Transformer | 0.98/1.62 | 0.82/1.35 |

[*] results reported from Xu et al. (2022b)

| **2** | $mean_{20}$ | $\min_{20}$ |
|---|---|---|
| GroupNet (Xu et al., 2022a) | 2.84/5.15 | 0.94/1.22 |
| LED (Mao et al., 2023) | 3.83/6.03 | 0.81/1.10 |
| MoFlow (Fu et al., 2025) | 2.38/4.61 | **0.71/0.86** |
| STFlow | **1.51/2.59** | 1.17/1.94 |
| STFlow-Transformer | 1.63/2.81 | 1.28/2.13 |

### 4.1.3. HUMAN TRAJECTORY FORECASTING

**Datasets and setup** Lastly, we also evaluate on the forecasting of basketball player trajectories from the SportVU NBA movement dataset (Yue et al., 2014), containing player moment from the 2015-2016 NBA season. Each data sample contains 10 player trajectories and 1 ball trajectory. We consider two separate evaluation setups used for this dataset, each with their own set of results from previous works. **1.** $c = 8$ (0.96s), $f = 12$ (1.44s) based on Xu et al. (2022b). **2.** $c = 10$ (2.0s), $f = 20$ (4.0s) from Xu et al. (2022a).

**Baselines** The baseline approaches consist of five human trajectory forecasting-specific methods and one general approach. For setup **1**: Trajectron++ (Salzmann et al., 2020), SVAE (Xu et al., 2022b); and LaM-SLidE (Sestak et al., 2025). Setup **2** contains three works from the CVPR community, namely GroupNet (Xu et al., 2022a), LED (Mao et al., 2023) and MoFlow (Fu et al., 2025).

**Metrics** Evaluation on the minimum ADE and minimum FDE of 20 inference runs per test sample is performed, as is common in human trajectory forecasting literature. For setup **2**, also the mean of 20 predictions in ADE and FDE is reported. The unit of measurement is feet in setup **1** and meters in setup **2**. In setup **1** the ball trajectory is not part of the prediction target while for **2** it is, to stay consistent with previous literature and implementations.

**Implementation** Two binary encodings are given as node features to indicate whether a trajectory is a ball and to which team the player belongs to. Two layers of STFlow are used. Other details are equal to the two other datasets, only 5 denoising steps are taken during inference.

*Table 3.* Conditional Generation of atom trajectories from MD17 dataset, averaged over 5 runs.

| | Aspirin | | Benzene | | Ethanol | | Malonaldehyde | | Naphthalene | | Salicylic | | Toluene | | Uracil | |
|---|---|---|---|---|---|---|---|---|---|---|---|---|---|---|---|---|
| | ADE | FDE | ADE | FDE | ADE | FDE | ADE | FDE | ADE | FDE | ADE | FDE | ADE | FDE | ADE | FDE |
| Prior $p_0$ | 0.513 | 0.848 | 0.285 | 0.459 | 0.706 | 1.207 | 0.502 | 0.831 | 0.468 | 0.728 | 0.474 | 0.755 | 0.540 | 0.840 | 0.462 | 0.737 |
| TFN[*] [(2018b)](#) | 0.133 | 0.268 | 0.024 | 0.049 | 0.201 | 0.414 | 0.184 | 0.386 | 0.072 | 0.098 | 0.115 | 0.223 | 0.090 | 0.150 | 0.090 | 0.159 |
| SE(3)-Tr[*] [(2020)](#) | 0.294 | 0.556 | 0.027 | 0.056 | 0.188 | 0.359 | 0.214 | 0.456 | 0.069 | 0.103 | 0.189 | 0.312 | 0.108 | 0.184 | 0.107 | 0.196 |
| EGNN[*] [(2021)](#) | 0.267 | 0.564 | 0.024 | 0.042 | 0.268 | 0.401 | 0.393 | 0.958 | 0.095 | 0.133 | 0.159 | 0.348 | 0.207 | 0.294 | 0.154 | 0.282 |
| EqMotion[*] [(2023)](#) | 0.185 | 0.246 | 0.029 | 0.043 | 0.152 | 0.247 | 0.155 | 0.249 | 0.073 | 0.092 | 0.110 | 0.151 | 0.097 | 0.129 | 0.088 | 0.116 |
| GeoTDM[*] [(2024)](#) | 0.107 | 0.193 | 0.023 | 0.039 | 0.115 | 0.209 | 0.107 | 0.176 | 0.064 | 0.087 | 0.083 | 0.120 | 0.083 | 0.121 | 0.074 | 0.099 |
| LaM-SLidE [(2025)](#) | **0.059** | **0.098** | 0.021 | 0.032 | 0.087 | 0.167 | 0.073 | 0.124 | 0.037 | 0.058 | 0.047 | 0.074 | 0.045 | 0.075 | 0.050 | **0.074** |
| STFlow | 0.076 | 0.142 | **0.011** | **0.018** | 0.078 | 0.144 | **0.051** | **0.093** | 0.025 | 0.041 | 0.041 | 0.068 | **0.036** | **0.060** | 0.050 | 0.079 |
| STFlow-Transformer | 0.070 | 0.132 | 0.012 | 0.019 | **0.050** | **0.095** | 0.058 | 0.111 | **0.025** | 0.044 | 0.042 | 0.073 | 0.045 | 0.077 | **0.050** | 0.082 |

[*] results reported from [Han et al. (2024)](#)

**Results** Looking at Table [4](#), we can see that STFlow achieves comparable to or slightly below relevant approaches in terms of the $\min_{20} ADE$ and $\min_{20} FDE$ metrics in setup **1**. The results from setup **2** display that the K-shot methods of LED ([Mao et al., 2023](#)) and MoFlow ([Fu et al., 2025](#)) perform well on the $\min_{20}$ metric, but seem to produce high mean error. We observe that STFlow in general produces relatively low mean errors, but higher $\min_{20}$ errors, indicating that the model prioritizes fidelity over diversity. Overall, evaluation on the human trajectory forecasting setting exhibits that STFlow yields competitive mean performance but higher $\min_{20}$ errors when comparing with state-of-the-art methods explicitly designed for this domain. Results in setup **2** are reproduced using the publicly released code and pretrained weights.

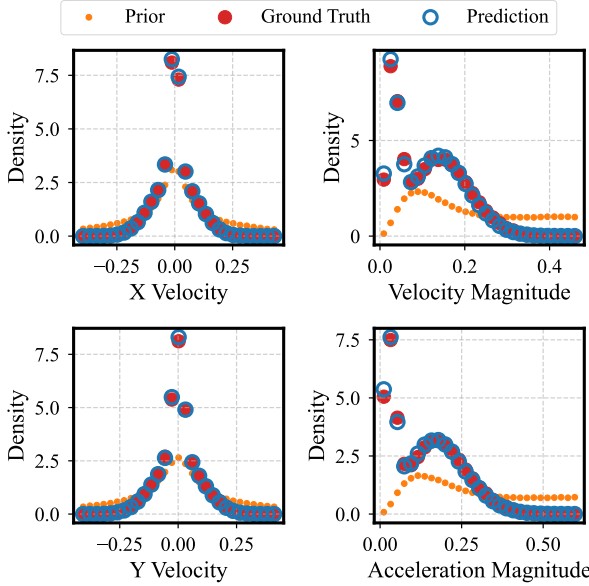

*Figure 3.* Velocity and acceleration density estimates from the 20 predicted and ground truth frames of the MD17 Ethanol test set ($n = 216320$)

### 4.2. Density evaluation

To assess whether STFlow accurately captures the distributional structure of trajectory dynamics beyond point-wise prediction errors, we compare the simulated and ground-truth densities of position-based features for the MD17 Ethanol test set results, shown in Figure [3](#). In particular, we examine velocity and acceleration densities, providing a diagnostic of the dynamics. Even though the uni-modal velocity and acceleration magnitude densities from the coupled prior $p_0$ (Eq. [2](#)) deviate substantially from the target densities, STFlow learns a mapping that recovers its multi-modal structure. As shown in Figure [4.2](#), the predicted velocity and acceleration densities closely match the shape, support and modal structure of the true distributions, demonstrating that the model effectively learns to correct the prior into realistic dynamics.

### 4.3. Scaling

In addition we analyze how STFlow scales in time and memory requirements with respect to the size of the system $\mathcal{G}$. The number of frames $T$ is set ranging from $T = 150$ to $T = 4000$ in the MD17 Aspirin dataset ($N = 21$), the largest publicly available setup provided by the state-of-the-art methods we compare with, GeoTDM ([Han et al., 2024](#)) and LaM-SLidE ([Sestak et al., 2025](#)). We measure both the average time per step (s) and maximum allocated GPU memory (GB) during training[1] with a batch size of 1 for all three models using equal hardware and software. The results, plotted on logarithmic scale in Figure [4](#), show that STFlow requires notably less time and GPU memory resources than the two state-of-the-art architectures for longer trajectory lengths. Quantitatively, on average a $1.9\times$ reduction in time per step and a $1.56\times$ reduction in memory requirements across all experiments. Measurements of GeoTDM are limited, as the three largest experiments did not fit into GPU memory (94GB).

---

[1]We measured only stage 2 from the LaM-SLidE ([Sestak et al., 2025](#)) approach, the most resource-demanding training stage.

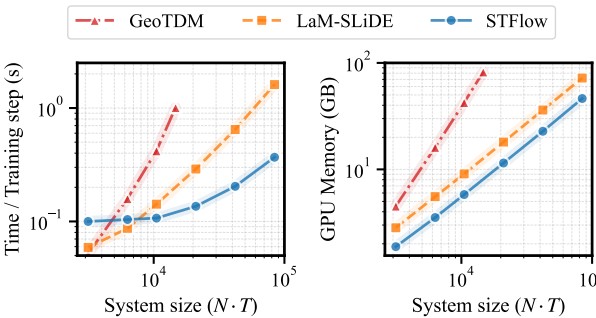

*Figure 4.* Scaling experiments on the MD17 Aspirin dataset. Time per training steps (s) and max. allocated GPU memory (GB) are measured for increasing trajectory length. Both axis are log-scaled.

*Table 5.* Ablations on prior and architecture, performed on the NBody Gravity dataset. Metric is FDE, averaged over 5 runs. Trained for 250 epochs in equal setting.

|  | UNet | Transformer | $\|u\|$ |
|---|---|---|---|
| STFlow (baseline) | 0.256 | **0.139** | 0.22 |
| **Prior** | | | |
| 1. Not Coupled $\zeta \sim \mathcal{N}(\mathbf{0}, \mathbf{I})$ | 0.382 | 0.307 | 1.51 |
| 2. Partly Coupled $\triangle\zeta \sim \mathcal{N}(\mathbf{0}, \mathbf{I})$ | 0.359 | 0.288 | 1.02 |
| 3. Coupled, $p_0$, $s = 1$ (2) | **0.252** | 0.231 | 0.13 |
| **Architecture** | | | |
| 4. w/o Spatial layer | 1.163 | 1.015 | |
| 5. w/o Temporal layer | 1.651 | 1.541 | |
| 6. w/o Edge updating | 0.272 | 0.201 | |

## 4.4. Ablation study

In order to quantify the contributions of the components of STFlow, we change individual parts and investigate the discrepancy in the FDE metric on the NBody Gravity dataset. Results can be seen in Table 5.

**Ablations on data-dependent coupling** Three different ablations are conducted on the informed prior $p_0$ to analyze the addition of data-dependent coupling. Namely, **1.** A Gaussian $\mathcal{N}(\mathbf{0}, \mathbf{I})$ on positions as prior, the standard in literature, **2.** A random walk starting at the last observed frame with velocities sampled from $\mathcal{N}(\mathbf{0}, \mathbf{I})$, **3.** Our informed random walk prior $p_0$ (Eq. 2), with the velocity variance scale factor set to $s = 1$ instead of $s = 4$. Additionally the estimated transport cost as the mean of the magnitude of the target vector field $u$ over the training data is shown on the right. When STFlow's data-dependent coupling approach is not used, prediction error increases by $85\%$ (**1**) and $74\%$ (**2**) on average. These priors (**1, 2**) have a higher associated transport cost, increasing complexity of the learning task. Ablations **1** and **2** demonstrate that the informed prior of the STFlow approach significantly improves simulation performance and that STFlow's construction of data-dependent couplings is highly effective in this setting. The parameter $s$ has more limited impact (**3**). In our experience it's influence is dataset-specific and $s$ can be used as hyperparameter for tuning the amount of noise in the prior.

**Ablations on model architecture** To demonstrate necessity of the Spatial and Temporal layers we remove them in experiments **4** and **5**. In ablation **4** nodes are unable to exchange information between other nodes across time and in ablation **5** across space respectively, drastically increasing prediction errors. As the last ablation (**6**), we let the learned edge representations from $\mathcal{E}$ unchanged after each layer instead of updating them. This results in $25\%$ higher error, showing that our approach benefits from iteratively updating its edge information during denoising.

## 5. Conclusion

We present STFlow, a novel flow matching approach designed for probabilistic simulation of geometric trajectories in N-body systems. By incorporating data-dependent couplings through an informed prior, STFlow learns more direct mappings from prior to target distributions. As a result of simplifying the modeling task and lowering the transport cost, STFlow yields consistent improvements in prediction accuracy across the domains of molecular dynamics, physical N-body systems and human trajectory forecasting. Additionally, by respecting fundamental permutation and time-shift symmetries inherent in N-body problems, STFlow demonstrates efficiency and scalability, allowing practical one-shot simulation of long temporal horizons.

Our experimental evaluation shows a mean decrease in prediction errors up to $56.9\%$ across benchmarks from multiple data domains while using only 5 function evaluations during simulation and maintaining linear scaling in both trajectory length and number of objects. These results demonstrate that the incorporation of both conditional and domain-specific information into the prior translates directly into both sample quality and computational efficiency, validating a principled approach to prior construction for conditional flow matching in trajectory simulation.

**Limitations and future work** Our approach is designed for equal-length trajectories per dataset. Decomposing them and using an autoregressive approach could be a viable approach for flexibility, alongside Neural Operators (Kovachki et al., 2023). Additionally, the usage of a random walk as prior might be unsuitable for systems with periodicity, strong external influences or hard constraints. In this case, more involved prior distributions can straightforwardly be integrated into STFlow, as long as sampling from such priors remains efficient. An analysis of the trade-off between diversity and fidelity within the prior construction is something we leave as future work. Finally, STFlow allows for generation of trajectories conditioned on arbitrary observed

timesteps, and investigating performance in applications of endpoint-conditioned simulation, super-resolution, or imputation, we see as promising future research.

## Impact Statement

This paper presents work whose goal is to advance the field of Machine Learning. Primary applications include simulation of physical and social systems. Beneficial uses include improving molecular simulations and physics-based models for scientific discovery in general. Potential concerns arise primarily in the human trajectory forecasting domain, where improved forecasting accuracy could enable more sophisticated surveillance.

We note that our method is domain-agnostic and does not introduce capabilities fundamentally different from existing trajectory prediction methods. We encourage users of this technology in a social setting to consider privacy implications and ensure compliance with relevant regulations.

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

# A. Technical Appendices and Supplementary Material

## A.1. Detailed training and inference procedures

The training and inference algorithms describing our approach are denoted below in Algorithm 1 and Algorithm 2.

---

**Algorithm 1** Conditional Flow Matching Training

---

**Require:** Train Dataset $p_1$, Model $v_\theta(\mathbf{x}, \mathbf{h}, \mathcal{E}, \tau)$
  **while** Training **do**
    $\mathbf{x}_1 \sim p_1$, $\tau \sim \mathcal{U}[0, 1]$               # Sample batch $\mathbf{x}_1$ and denoising level $\tau$
    $\mathbf{x}_0^{0:c} \leftarrow \mathbf{x}_1^{0:c}$, $\mathbf{x}_0^{c:T} \leftarrow \zeta(\mathbf{x}_1^{0:c})$      # Make informed prior $\mathbf{x}_0$ from observed frames $\mathbf{x}_1^{0:c}$
    $\mathbf{x}_\tau \sim p_\tau = \mathcal{N}(\tau \mathbf{x}_1 + (1-\tau)\mathbf{x}_0, \sigma_p^2)$    # Sample trajectories from Gaussian probability path
    $u \leftarrow \mathbf{x}_1 - \mathbf{x}_0$                   # Calculate true vector field
    $\mathbf{h} \leftarrow \text{Features}(\mathbf{x}_\tau)$, $\mathcal{E} \leftarrow \text{Connectivity}(\mathbf{x}_\tau)$     # Construct geometric trajectory
    $v_\theta \leftarrow v_\theta(\mathbf{x}_\tau, \mathbf{h}, \mathcal{E}, \tau)$                  # Forward pass
    $\mathcal{L}_{CFM}(\theta) \leftarrow \|v_\theta - u\|^2$              # Calculate loss
    $\theta \leftarrow \theta + \alpha \nabla_\theta \mathcal{L}_{CFM}(\theta)$           # Backpropagate and optimize
  **end while**

---

---

**Algorithm 2** Inference using Euler Integration

---

**Require:** Test Dataset $p_1$, Trained model $v_\theta(\mathbf{x}, \mathbf{h}, \mathcal{E}, \tau)$, Number of Function Evaluations $n$
  **for** $\mathbf{x}_1 \in p_1$ **do**
    $\mathbf{x}_0^{0:c} \leftarrow \mathbf{x}_1^{0:c}$, $\mathbf{x}_0^{c:T} \leftarrow \zeta(\mathbf{x}_1^{0:c})$      # Make informed prior $\mathbf{x}_0$ from observed frames $\mathbf{x}_1^{0:c}$
    $\hat{\mathbf{x}} \leftarrow \mathbf{x}_0$, $\Delta\tau \leftarrow \frac{1}{n}$
    **for** $\tau \in \{0, \frac{1}{n}, ..., \frac{n-1}{n}\}$ **do**
      $\mathbf{h} \leftarrow \text{Features}(\hat{\mathbf{x}})$, $\mathcal{E} \leftarrow \text{Connectivity}(\hat{\mathbf{x}})$     # Construct geometric trajectory
      $v_\theta \leftarrow v_\theta(\hat{\mathbf{x}}, \mathbf{h}, \mathcal{E}, \tau)$             # Forward pass
      $\hat{\mathbf{x}} \leftarrow \hat{\mathbf{x}} + \Delta\tau \cdot v_\theta$            # Update $\hat{\mathbf{x}}$ using Euler method
    **end for**
  **end for**

---

## A.2. Definitions of EGCL and the UNet

Our Spatial Message Passing layer is based on the Equivariant Graph Convolutional Layer (Satorras et al., 2021), which we define as:

$$\mathbf{m}_{ij} = \varphi_{\mathbf{m}}(\mathbf{h}_i, \mathbf{h}_j, \mathbf{e}_{ij}, \|\mathbf{x}_i - \mathbf{x}_j\|^2, \tau_{\text{emb}}, t_{\text{emb}}),$$

$$\mathbf{h}_i^{'} = \text{LayerNorm}\left(\varphi_{\mathbf{h}}\left(\mathbf{h}_i, \tau_{\text{emb}}, \sum_{j \in \mathcal{N}(i)} \mathbf{m}_{ij}\right) + \mathbf{h}_i\right),$$

$$\mathbf{x}_i^{'} = \mathbf{x}_i + \frac{1}{|\mathcal{N}(i)|} \sum_{j \in \mathcal{N}(i)} \varphi_{\mathbf{x}}(\mathbf{m}_{ij})(\mathbf{x}_i - \mathbf{x}_j),$$

where $\varphi_{\mathbf{m}}, \varphi_{\mathbf{h}}, \varphi_{\mathbf{x}}$ are 2-layer MLPs with SiLU activation functions and $\tau_{\text{emb}}, t_{\text{emb}}$ are sinusiodal embeddings with 16 dimensions based on $\tau$ and timestep number $t$.

Our Temporal Convolution layer is based on the UNet design from (Dhariwal & Nichol, 2021), for which we use the following parameters in Table 6. The input of the UNet consists of a batch of $\mathbf{h}$ and a sinusoidal embedding of $\tau$ of 16 dimensions. As our benchmarks experimentally seem to contain dependence on a notion of time, a sinusoidal frame index embedding is also added to $\mathbf{h}$. We convolve over the timestep axis, containing $T$ frames for each trajectory. The output is the new $\mathbf{h}$ and the change in the velocity vector field, $\Delta\mathbf{v}$, calculated using a 2-layer MLP $\varphi_{\mathbf{v}}(\mathbf{h})$.

*Table 6.* UNet parameters used in the Temporal Convolution layer

| parameter | value |
|---|---|
| image_size | $T$ |
| dims | 1 |
| in_channels | $h = 64$ |
| model_channels | $h = 64$ |
| out_channels | $h = 64$ |
| num_res_blocks | 2 |
| channel_mult | (1,2) |
| num_heads | 2 |
| num_head_channels | $h = 64$ |
| use_scale_shift_norm | True |
| resblock_updown | True |
| use_new_attention_order | True |
| attention_resolutions | [] |

### A.3. Hyperparameters

The hyperparameters used to train and evaluate our model on every dataset are included in Table 8. As optimizer we used AdamW with default parameters in combination with a ReduceLROnPlateau based on minimum validation loss with factor 0.5 and a patience of 30 or 50. In Table 8, we specify connectivity, $s$ is the parameter determining the amount of extra noise added in the informed prior by scaling the used variance of the velocities of the observed frames. We did not perform an automated hyperparameter search for optimal configurations, so improvements can be made with hyperparameter optimization. We list both training hyperparameters setups for UNet and Transformer experiments.

*Table 7.* Training hyperparameters of all conditional generation experiments

| | connectivity | #augmentations | #layers | $\tau$ distribution | $s$ | lr | epochs | val/test | batch size | hidden dim |
|---|---|---|---|---|---|---|---|---|---|---|
| **N-body** | | | | | | | | | | |
| Gravity | full | 12 | 3 | $\sqrt{\mathcal{U}[0,1]}$ | 4 | $5 \cdot 10^{-4}$ | 400 | 0.15/0.15 | 32 | 64 |
| Springs | full | 8 | 2 | $\sqrt{\mathcal{U}[0,1]}$ | 4 | $5 \cdot 10^{-4}$ | 300 | 0.15/0.15 | 32 | 64 |
| Charged | full | 14 | 3 | $\sqrt{\mathcal{U}[0,1]}$ | 4 | $5 \cdot 10^{-4}$ | 400 | 0.15/0.15 | 32 | 64 |
| **NBA** | | | | | | | | | | |
| Rebound | full | 4 | 2 | $\sqrt{\mathcal{U}[0,1]}$ | 4 | $5 \cdot 10^{-4}$ | 500 | 0.15 | 64 | 64 |
| Score | full | 0 | 2 | $\sqrt{\mathcal{U}[0,1]}$ | 4 | $5 \cdot 10^{-4}$ | 500 | 0.15 | 64 | 64 |
| Secondary setup | full | 3 | 2 | $\sqrt{\mathcal{U}[0,1]}$ | 4 | $5 \cdot 10^{-4}$ | 300 | 0.1 | 64 | 64 |
| **MD17** | | | | | | | | | | |
| Aspirin | full | 12 | 3 | $\sqrt{\mathcal{U}[0,1]}$ | 4 | $5 \cdot 10^{-4}$ | 400 | 0.15/0.15 | 32 | 64 |
| Benzene | full | 5 | 3 | $\sqrt{\mathcal{U}[0,1]}$ | 4 | $5 \cdot 10^{-4}$ | 400 | 0.15/0.15 | 32 | 64 |
| Ethanol | full | 6 | 3 | $\sqrt{\mathcal{U}[0,1]}$ | 4 | $5 \cdot 10^{-4}$ | 400 | 0.15/0.15 | 32 | 64 |
| Malonaldehyde | full | 6 | 3 | $\sqrt{\mathcal{U}[0,1]}$ | 2 | $5 \cdot 10^{-4}$ | 400 | 0.15/0.15 | 32 | 64 |
| Naphthalene | full | 6 | 3 | $\sqrt{\mathcal{U}[0,1]}$ | 4 | $5 \cdot 10^{-4}$ | 400 | 0.15/0.15 | 32 | 64 |
| Salicylic | full | 6 | 3 | $\sqrt{\mathcal{U}[0,1]}$ | 4 | $5 \cdot 10^{-4}$ | 400 | 0.15/0.15 | 32 | 64 |
| Toluene | full | 4 | 3 | $\sqrt{\mathcal{U}[0,1]}$ | 4 | $5 \cdot 10^{-4}$ | 400 | 0.15/0.15 | 32 | 64 |
| Uracil | full | 12 | 3 | $\sqrt{\mathcal{U}[0,1]}$ | 4 | $5 \cdot 10^{-4}$ | 400 | 0.15/0.15 | 32 | 64 |

### A.4. Compute resources

Most experiments were run using 1 GeForce RTX 3080 (10GB) or 1 Tesla V100 (16GB). However, some scaling and transformer experiments required more GPU memory, in which case 1 A100 (40GB) or 1 H100 (95GB) was used. Training the models took in the range from 8 (Springs) to 41 hours (NBA Score), while all inference runs that were performed either 5 or 20 times on each test set sample too a few minutes to an hour at most. The total runtime of all 40 training experiments consists of roughly 750 GPU hours.

### A.5. Standard deviations

The means and standard deviations calculated from the 5 runs for each sample in the N-body and MD17 dataset and of the 20 runs for the NBA dataset are denoted in Table 9. The standard deviation here measures diversity, e.g. to which extent different initializations of the prior $p_0$ have an impact on STFlow's performance.

*Table 8.* Training hyperparameters of all conditional generation experiments using a Transformer as temporal layer

| | connectivity | #augmentations | #layers | $\tau$ distribution | $s$ | lr | epochs | val/test | batch size | hidden dim |
|---|---|---|---|---|---|---|---|---|---|---|
| **N-body** | | | | | | | | | | |
| Gravity | full | 12 | 3 | $\sqrt{\mathcal{U}[0,1]}$ | 4 | $5 \cdot 10^{-4}$ | 300 | 0.15/0.15 | 32 | 128 |
| Springs | full | 8 | 2 | $\sqrt{\mathcal{U}[0,1]}$ | 4 | $5 \cdot 10^{-4}$ | 300 | 0.15/0.15 | 32 | 128 |
| Charged | full | 14 | 3 | $\sqrt{\mathcal{U}[0,1]}$ | 4 | $5 \cdot 10^{-4}$ | 400 | 0.15/0.15 | 32 | 128 |
| **NBA** | | | | | | | | | | |
| Rebound | full | 4 | 2 | $\sqrt{\mathcal{U}[0,1]}$ | 4 | $5 \cdot 10^{-4}$ | 250 | 0.15 | 64 | 128 |
| Score | full | 0 | 2 | $\sqrt{\mathcal{U}[0,1]}$ | 4 | $5 \cdot 10^{-4}$ | 300 | 0.15 | 64 | 128 |
| Secondary setup | full | 3 | 2 | $\sqrt{\mathcal{U}[0,1]}$ | 4 | $5 \cdot 10^{-4}$ | 300 | 0.1 | 64 | 128 |
| **MD17** | | | | | | | | | | |
| Aspirin | full | 12 | 3 | $\sqrt{\mathcal{U}[0,1]}$ | 4 | $5 \cdot 10^{-4}$ | 400 | 0.15/0.15 | 32 | 128 |
| Benzene | full | 5 | 3 | $\sqrt{\mathcal{U}[0,1]}$ | 4 | $5 \cdot 10^{-4}$ | 400 | 0.15/0.15 | 32 | 128 |
| Ethanol | full | 6 | 3 | $\sqrt{\mathcal{U}[0,1]}$ | 4 | $5 \cdot 10^{-4}$ | 400 | 0.15/0.15 | 32 | 128 |
| Malonaldehyde | full | 6 | 3 | $\sqrt{\mathcal{U}[0,1]}$ | 2 | $5 \cdot 10^{-4}$ | 400 | 0.15/0.15 | 32 | 128 |
| Naphthalene | full | 6 | 3 | $\sqrt{\mathcal{U}[0,1]}$ | 4 | $5 \cdot 10^{-4}$ | 400 | 0.15/0.15 | 32 | 128 |
| Salicylic | full | 6 | 3 | $\sqrt{\mathcal{U}[0,1]}$ | 4 | $5 \cdot 10^{-4}$ | 400 | 0.15/0.15 | 32 | 128 |
| Toluene | full | 4 | 3 | $\sqrt{\mathcal{U}[0,1]}$ | 4 | $5 \cdot 10^{-4}$ | 400 | 0.15/0.15 | 32 | 128 |
| Uracil | full | 12 | 3 | $\sqrt{\mathcal{U}[0,1]}$ | 4 | $5 \cdot 10^{-4}$ | 400 | 0.15/0.15 | 32 | 128 |

*Table 9.* Mean and standard deviation across 5 or 20 samples of ADE and FDE from evaluating STFlow across the NBody, NBA and MD17 datasets

| | # Simulations | ADE | FDE |
|---|---|---|---|
| NBody Charged | 5 | $0.085 \pm 0.012$ | $0.158 \pm 0.029$ |
| NBody Springs | 5 | $0.0004 \pm 0.00005$ | $0.0008 \pm 0.00009$ |
| NBody Gravity | 5 | $0.107 \pm 0.011$ | $0.231 \pm 0.030$ |
| NBA Setup 2 | 20 | $1.51 \pm 0.19$ | $2.59 \pm 0.37$ |
| MD17 Aspirin | 5 | $0.076 \pm 0.004$ | $0.142 \pm 0.010$ |
| MD17 Benzene | 5 | $0.011 \pm 0.001$ | $0.018 \pm 0.001$ |
| MD17 Ethanol | 5 | $0.078 \pm 0.001$ | $0.144 \pm 0.001$ |
| MD17 Naphthalene | 5 | $0.025 \pm 0.001$ | $0.041 \pm 0.003$ |
| MD17 Salicylic | 5 | $0.041 \pm 0.003$ | $0.068 \pm 0.007$ |
| MD17 Toluene | 5 | $0.036 \pm 0.003$ | $0.060 \pm 0.006$ |
| MD17 Uracil | 5 | $0.050 \pm 0.002$ | $0.079 \pm 0.005$ |

## A.6. Additional plots

We provide additional plots of trajectories from the N-body and MD17 datasets in the Figures below. As well as additional density evaluation on the Gravity and NBA Score test sets.

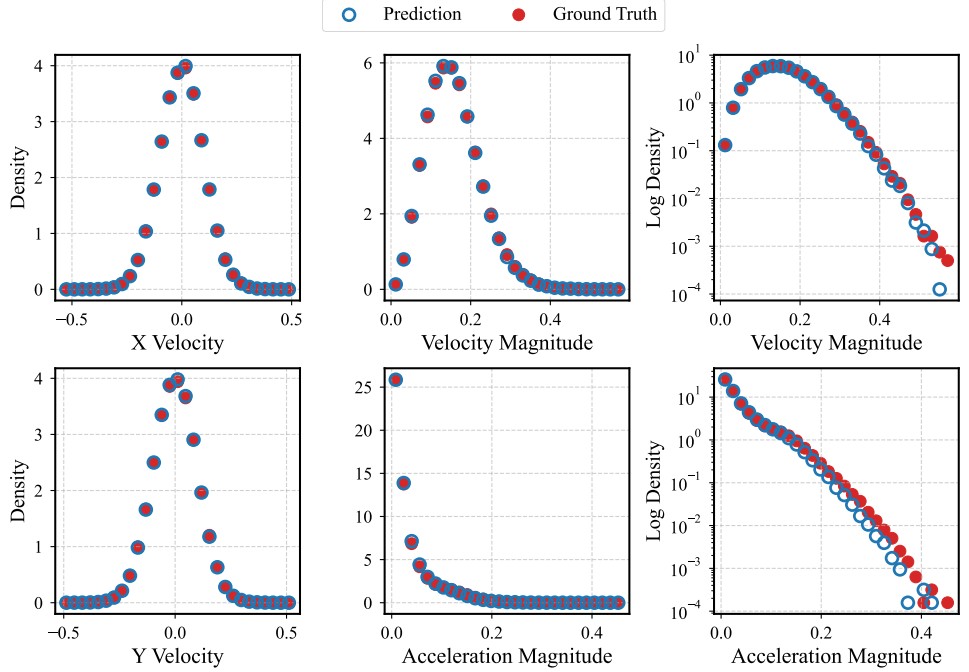

*Figure 5.* Density evaluation of velocities and accelerations from the Gravity test set inference trajectories

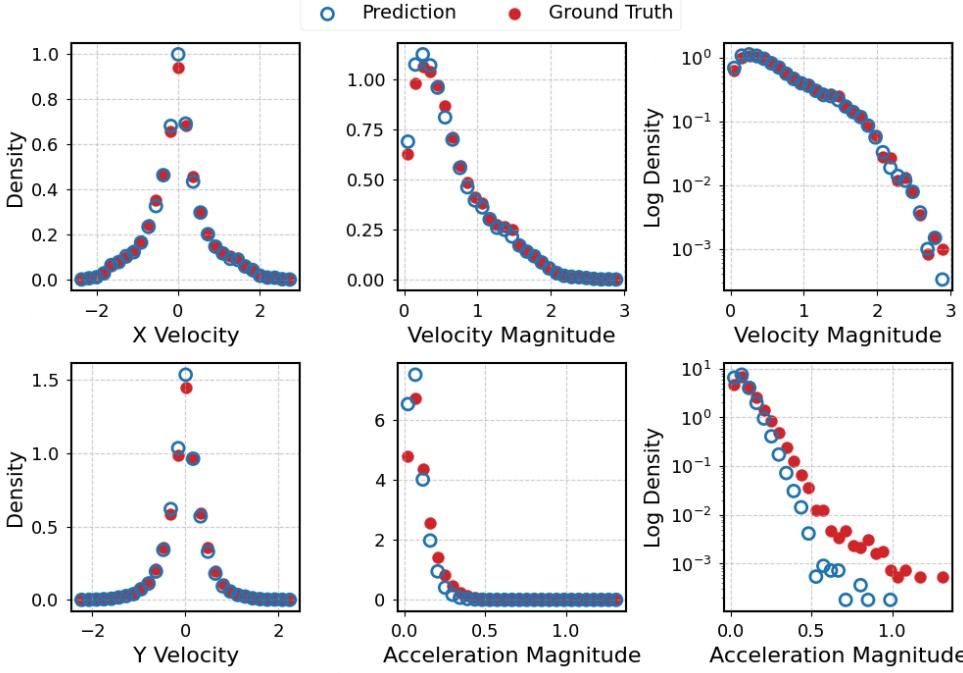

*Figure 6.* Density evaluation of velocities and accelerations from the NBA Score test set inference trajectories

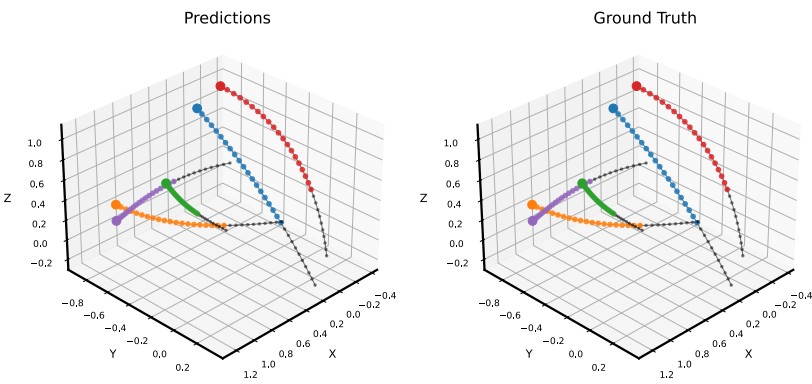

*Figure 7.* Trajectory inference results from the N-body Springs test set

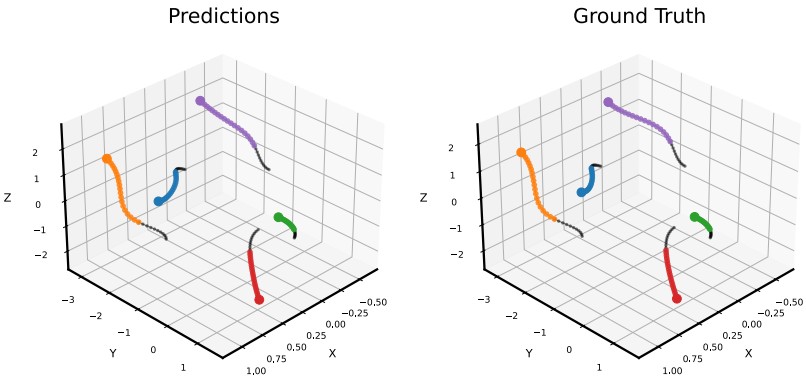

*Figure 8.* Trajectory inference results from the N-body Charged test set

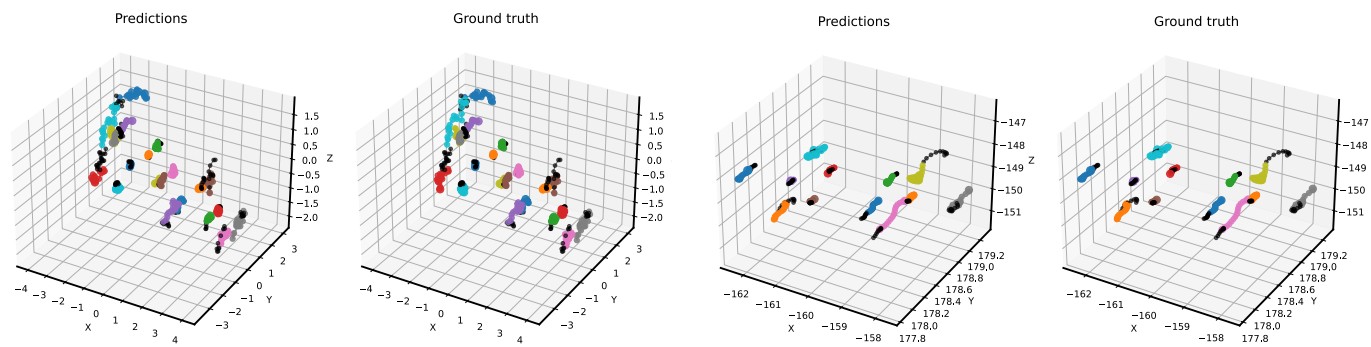

*Figure 9.* Aspirin (left) and Benzene (right) trajectory inference results from the MD17 test set

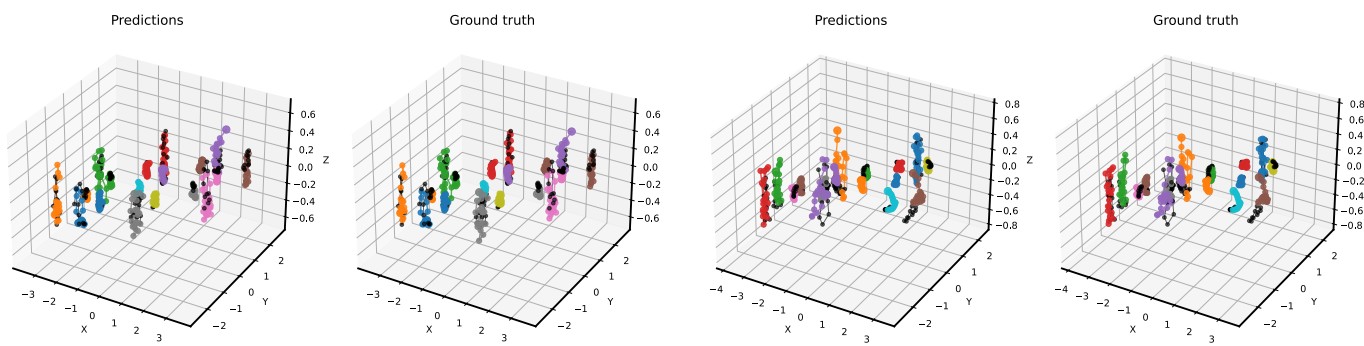

*Figure 10.* Ethanol (left) and Malonaldehyde (right) trajectory inference results from the MD17 test set

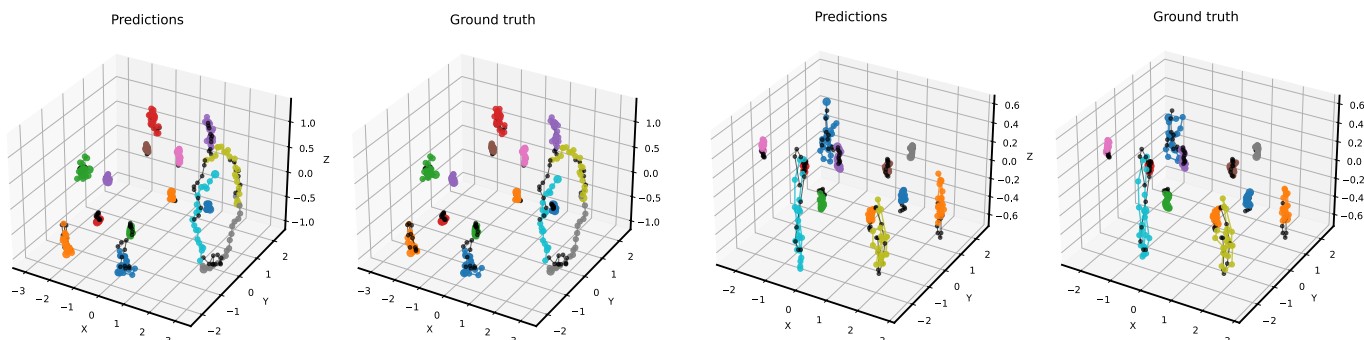

*Figure 11.* Naphthalene (left) and Salicylic (right) trajectory inference results from the MD17 test set

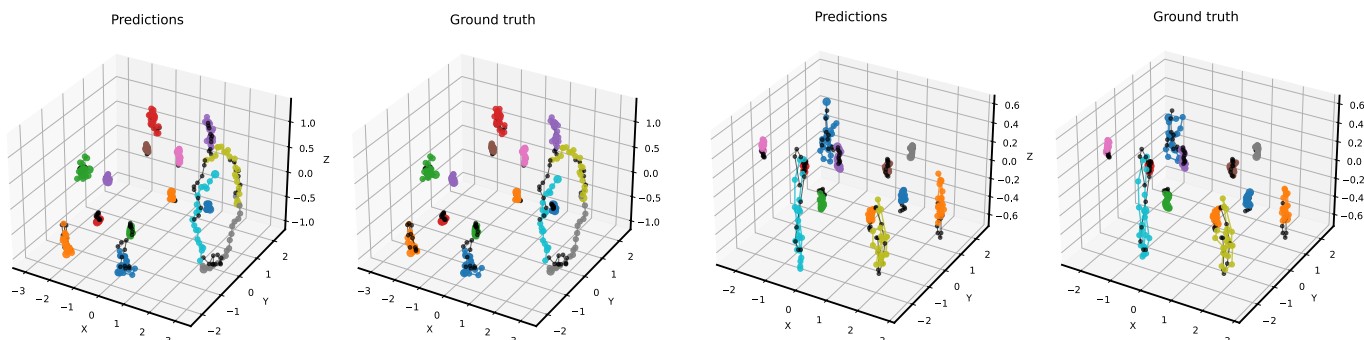

*Figure 12.* Toluene (left) and Uracil (right) trajectory inference results from the MD17 test set

