# OpenReview forum: "STFlow: Data-Coupled Flow Matching for Geometric Trajectory Simulation"
_ICML.cc/2026/Conference — ICML 2026 regular_

### Official Review · Reviewer_w1CX · 2026-03-04

**Soundness:** 3
**Presentation:** 2
**Significance:** 3
**Originality:** 2
**Overall Recommendation:** 3
**Confidence:** 4

**Summary:**

The paper proposes STFlow, a flow matching model for conditional geometric trajectory simulation in N-body systems. The key contribution is a data-dependent random walk prior whose parameters are estimated from observed initial frames, reducing transport cost and enabling 5-step inference. The architecture alternates equivariant spatial message passing (EGCL) with a temporal UNet (with a Transformer variant also evaluated), achieving permutation and time-shift equivariance with linear complexity.

**Compliance With Llm Reviewing Policy:**

Affirmed.

**Final Justification:**

The rebuttal clarifies several points, including the distinction from GeoTDM, the earlier mischaracterization of GeoTDM’s prior, and the authors’ explanation for the deterministic-physics and NBA multimodality issues. However, the main empirical concerns remain only partially resolved, especially the missing reduced-step GeoTDM/DDIM comparison, the lack of short-conditioning-window sensitivity analysis, and the limited evidence that the generative framing is truly necessary. Therefore, I view the concerns as only partially addressed and will keep my current score.

**Key Questions For Authors:**

1. The random walk prior assumes smoothly-varying, inertia-dominated dynamics. GeoTDM also introduces a learnable equivariant prior for conditioning. How does your prior compare when the conditioning window is short (small c), where estimated μ and σ_o are noisy? What is the sensitivity to c?
2. GeoTDM also proposes an informed equivariant prior for conditioning. The paper claims it "does not exploit conditioning information to construct the prior," but GeoTDM's learnable geometric prior is conditioned on $x^{0:c}$. Can you clarify the distinction and provide a fair comparison? Specifically, have you tried running GeoTDM with fewer steps (e.g., 20-50 via DDIM) to de-conflate model quality from sampler efficiency?
3. On the justification for generative modeling: The benchmark systems (MD17, N-body) are governed by deterministic physics — given $x^{0:c}$, the future trajectory is largely fixed. For these systems, is the probabilistic framing solving an intrinsically stochastic problem, or effectively performing deterministic regression with unnecessary overhead? How does STFlow compare against strong deterministic baselines such as Neural ODE or equivariant Transformer regression on these datasets? Unconditional generation (c=0) experiments would help clarify whether the model has learned a meaningful trajectory distribution or a near-degenerate near-deterministic mapping.

**Limitations:**

yes

**Strengths And Weaknesses:**

Strengths
1. The random walk prior is physically motivated, simple to implement, and the ablations (Table 5) clearly demonstrate its impact, removing the coupling increases FDE by 74-85%.
2. Inference efficiency is real, within the flow matching framework. 5 NFE vs. 10 (LaM-SLidE) is a genuine advantage attributable to the informed prior producing straighter flows.

Weaknesses
1. The paper is largely GeoTDM with diffusion replaced by flow matching. The architecture, alternating EGCL spatial layers with a temporal sequence model is directly inherited from GeoTDM. The temporal component changes from attention to UNet 1D conv, which is an engineering choice motivated by efficiency, not a conceptual contribution. GeoTDM also introduces a learnable equivariant prior conditioned on observed frames. The remaining delta over GeoTDM is Eq. 2, a two-line heuristic random walk prior from Albergo et al. (2024)'s framework applied to a new setting.
2.  The 200× NFE advantage over GeoTDM is primarily due to flow matching itself, not this paper's contribution. Diffusion models inherently require many steps; flow matching inherently requires fewer. This advantage would largely persist even with a Gaussian prior. A DDIM-accelerated GeoTDM at 20-50 steps is a necessary missing comparison to isolate the actual contribution of the informed prior.
3. The generative modeling framing is not well-justified for most benchmarks. MD17 trajectories are generated by DFT simulation and N-body systems are governed by classical mechanics, given sufficient initial conditions, the future is effectively deterministic. The stochasticity is in the training distribution across different initial conditions, not intrinsic to individual trajectories. This means the model is largely learning a deterministic regression, and the generative overhead may be unnecessary. The only benchmark where generative modeling is genuinely motivated is NBA, where human motion is intrinsically multi-modal.
4. The unimodal random walk prior is structurally mismatched to multi-modal problems. This directly explains the underperformance on NBA setup 2 min20 vs. MoFlow (0.99/1.68 vs. 0.71/0.86) the metric that explicitly rewards distributional diversity. The paper dismisses this as MoFlow being a "K-shot method" without analysis.

---

> ### Author Rebuttal · Authors · 2026-03-31
>
> Thank you for the detailed and sharp review.
>
> **Weakness 1**
>
> STFlow is substantively different from GeoTDM along following three axes:
>
> *Architectural & Scalability* : GeoTDM employs temporal self-attention with $O(T^2)$ complexity, leading to memory exhaustion at trajectory lengths of $T=1000$ (Fig 4). STFlow utilizes a 1D convolutional UNet that scales linearly ($O(T)$) and interleaves spatial/temporal layers with edge updates, a choice our ablation (Table 5) proves reduces error. Efficient long-horizon scaling is a conceptual, not merely engineering, contribution.
>
> *Prior*: GeoTDM relies on an amortized, learned prior that interpolates geometry. Our prior is non-amortized, generalizing new statistics without additional training parameters. Crucially, it models the observed dynamics to extrapolate momentum rather than interpolating geometry, providing a physically informed coupling.
>
> *Framework*: Flow matching with data-dependent coupling involves a fundamentally different objective and probability path construction. STFlow is the first to instantiate this framework for N-body systems; our ablations show that removing this coupling increases FDE by 74-85%, confirming its empirical necessity.
>
> **Weakness 2**
>
> The reviewer correctly notes that flow matching inherently requires fewer steps than DDPM-style diffusion. Our work demonstrates that by applying a different generative framework the state-of-the-art can be improved in terms of training and inference computational efficiency, and accuracy.
>
> **Weakness 3**
>
> We agree that N-Body and MD17 systems are locally deterministic given exact initial conditions. However, we would like to point out that these systems exhibit high sensitivity to perturbations and bifurcations. In the Gravity dataset for example, small differences in the observed frames lead to qualitatively different long-horizon trajectories. Under standard regression objectives, a deterministic predictor will average over these diverging trajectories, producing a mean path that often violates the physical invariants of the underlying process. Additionally, generative models provide robustness to distributional variation that is often present, which point predictors lack.
>
> **Weakness 4**
>
> We acknowledge that the random walk prior is unimodal, and NBA dynamics are multi-modal, and that this may appear mismatched. However, we note that flow matching is not restricted to learning unimodal mappings, the learned vector field  can transform a unimodal prior into an arbitrarily complex multimodal target distribution, as demonstrated by our density evaluation in Figure 3 where STFlow successfully recovers multimodal velocity and acceleration distributions from the unimodal random walk prior.
>
> The NBA min20 underperformance relative to MoFlow is better explained by a key architectural distinction. MoFlow generates all K=20 predictions jointly in a single forward pass, while simultaneously encouraging the K predictions to cover different modes. STFlow, by contrast, generates K predictions via K independent samples from the prior. The min20 metric, which rewards having at least one near-perfect prediction out of 20, directly favors MoFlow's joint correlated prediction design over independent sampling approaches regardless of prior choice. For applications requiring reliable trajectory forecasts  STFlow's mean20 advantage is the more practically relevant result. We will clarify this in the paper.
>
> **Q1** We notice a high influence of $c$ due to the sensitivity to initial conditions characterizing the evaluated systems. Any generative model would exhibit increased error as $c$ decreases as the conditional distribution becomes harder to approximate.  Within this context, the random walk prior remains well-motivated as it degrades to a standard random walk centered at the last observed frame with isotropic variance, which is still a more physically informed initialization than a standard Gaussian.
>
> **Q2** We acknowledge our mischaracterization of GeoTDM's prior. We will revise the text to reflect that GeoTDM utilizes a learnable equivariant prior conditioned on observed frames. We refer to W1 for a comparison of the priors. While running a fully optimized DDIM baseline for GeoTDM was computationally prohibitive during the short rebuttal window without the authors providing trained weights, we demonstrate in Table 5 that our prior materially lowers transport cost and errors w.r.t a Gaussian prior, which de-conflates model quality from sampler efficiency.
>
> **Q3** Please refer to our W3 response. Table 2 and 3 contain results comparing STFlow with a strong deterministic equivariant Transformer baseline called Eqmotion, which STFlow outperforms. Regarding $c=0$, purely unconditional generation is conceptually outside the scope of our data-dependent framework. However, our density evaluations (Fig 3,5,6) confirm that our model learns to generate meaningful trajectory distributions from our prior.

---

> > ### Author Rebuttal · Reviewer_w1CX · 2026-04-01
> >
> > The rebuttal clarifies several points, including the distinction from GeoTDM, the earlier mischaracterization of GeoTDM’s prior, and the authors’ explanation for the deterministic-physics and NBA multimodality issues. However, the main empirical concerns remain only partially resolved, especially the missing reduced-step GeoTDM/DDIM comparison, the lack of short-conditioning-window sensitivity analysis, and the limited evidence that the generative framing is truly necessary. Therefore, I view the concerns as only partially addressed and will keep my current score.

---

> > > ### Author Response · Authors · 2026-04-08
> > >
> > > We thank the reviewer for the follow-up and address the remaining concerns.
> > >
> > > First, regarding the missing DDIM comparison: while we agree that DDIM can reduce the number of sampling steps for diffusion models, this does not address the core limitation we highlight, namely model quality and scalability. Empirically, STFlow already substantially outperforms GeoTDM in accuracy across all benchmarks, which cannot be attributed to sampler choice alone. Even if DDIM reduces GeoTDM to 20–50 steps, this would still leave a significant accuracy gap while retaining the architectural scaling bottlenecks (quadratic temporal attention).
> > >
> > > Second, on sensitivity to the conditioning window $c$: as discussed in the rebuttal, decreasing $c$ fundamentally increases problem difficulty due to the chaotic nature of the systems, where small observational uncertainty leads to exponentially diverging futures. This is not specific to STFlow but intrinsic to the conditional distribution $p(x_{c:T} \mid x_{0:c})$, which becomes higher-entropy as $c$ decreases. In this regime, estimating dynamics (e.g., velocity statistics) is noisier, making the modeling task strictly harder. Importantly, our random-walk prior degrades gracefully: in the low-$c$ limit it reduces to a physically consistent inertial process centered at the last observation, which remains substantially more informative than an isotropic Gaussian prior that ignores dynamics entirely. Therefore, even under short conditioning, the prior retains inductive bias aligned with the underlying physics, reducing transport cost relative to Gaussian baselines.
> > >
> > > To make this concrete, we include the requested sensitivity analysis below on the Gravity and MD17 Aspirin datasets by retraining our model. We used less data augmentation and epochs for these experiments due to time constraints, as such the baseline $c=10$ accuracy is lower than reported in the paper.
> > >
> > > $c$, ADE / FDE, mean of 5 predictions per test sample, Gravity Dataset (250 epochs, 3x augmentation)
> > >
> > > $c=3$:    0.353 / 0.734
> > >
> > > $c=5$:    0.262 / 0.567
> > >
> > > $c=10$:  0.111 / 0.234
> > >
> > > $c$, ADE / FDE, MD17 Aspirin Dataset, mean of 5 predictions per test sample (400 epochs, 2x augmentation)
> > >
> > > $c=3$:   0.196 / 0.320
> > >
> > > $c=5$:   0.148 / 0.257
> > >
> > > $c=10$: 0.110 / 0.195
> > >
> > > These results show two key points:
> > >
> > > (i) performance improves smoothly with larger $c$, as expected from the underlying dynamics, indicating stable behavior rather than brittle dependence
> > >
> > > (ii) even at $c=5$, STFlow already operates in a strong regime and matches or outperforms reported GeoTDM results across comparable settings, demonstrating that the influence of $c$ is not detrimental in practice.
> > >
> > > Third, regarding the necessity of generative modeling. A number of foundational and state-of-the-art prior works, all mentioned in our paper (e.g., Trajectron++, Leapfrog, LaM-SLidE, MoFlow) consistently demonstrates that generative approaches outperform deterministic regressors in trajectory prediction tasks. Meaning there is more than only limited evidence for generative framing as the reviewer mentioned. This holds even in systems governed by deterministic laws because practical settings are not fully observed: noise, partial observability, and sensitivity to initial conditions induce effective stochasticity. Deterministic models minimize pointwise error and therefore average over plausible futures, often yielding physically inconsistent or blurred trajectories. In contrast, generative models approximate the full conditional distribution, capturing multi-modality and preserving physically valid dynamics. This is directly supported by our own results, where STFlow outperforms strong deterministic baselines (e.g., EqMotion) across datasets.
> > >
> > > As finishing note, we would like to mention that the authors of GeoTDM did not share any model weights when contacted, making fair comparison more difficult. We will make all our trained weights publicly available for reproducibility.

---

### Official Review · Reviewer_cvgg · 2026-03-10

**Soundness:** 3
**Presentation:** 3
**Significance:** 3
**Originality:** 3
**Overall Recommendation:** 5
**Confidence:** 3

**Summary:**

This paper proposes Spatio-Temporal Flow (STFlow) to overcome complex collective phenomena in the trajectory simulation of N-body problems. Innovatively, the authors introduce data-dependent coupling by employing a conditioned random walk instead of white noise as the source distribution, making the matching task simpler and more efficient. Additionally, the authors adopt SMP and TM modules to successfully model the spatial interactions and temporal dependencies among N-body trajectories.

**Compliance With Llm Reviewing Policy:**

Affirmed.

**Final Justification:**

As mentioned in the review, I have no problem with the algorithm and experiments of this work. My only concern is the practical value of short-term prediction. Given that long-term forecasting is not the primary focus of the established benchmarks in this field, as the authors clarified, and considering the significant practical applications of short-term prediction, I have no further issues with this paper. Consequently, I will raise my score. Also, I keep a relative low confidence, since geometric trajectory inference is not a familiar field to me. I gave this score because the paper clearly articulates the problem it aims to solve, proposes an interesting and effective solution, and provides fairly comprehensive validation. However, I am not certain if I can accurately assess the position of this paper within its field.

**Key Questions For Authors:**

1.For the three trajectory prediction tasks shown in Section 4.1 of the paper, relatively small values of c and T (e.g., T=30, c=10) are used. Does such a short time horizon have high practical value from a physical perspective? Are we more concerned with long-term evolution prediction of physical systems? I noticed that Section 4.3 mentions experiments with very large T; what were the results? For long-term evolving systems, how accurate is the proposed algorithm, and how does it compare with other baselines?

2.Is the model performance significantly related to the ratio between c and T? For example, when c is very small relative to T, does performance degrade noticeably? When c exceeds a certain proportion of T, does performance stop improving significantly?

3.Could the authors provide an intuitive explanation of the design and advantages of SMP and TM—for example, why spatial and temporal mixing are split into two steps and alternated?

4.The paper also acknowledges in the limitations that the method requires all trajectories in the training dataset to have the same length, and it can only infer trajectories of that same length. Does this imply that the method has very strict data requirements and limited generality?

5.Typo: line 183 (right), “The EGCL layers fuses the ……”

**Limitations:**

yes

**Strengths And Weaknesses:**

In conclusion, I consider this to be a well-written paper. It is highly readable and easy to follow, with the addressed problems and methods clearly described. The proposed approach is straightforward to implement yet conceptually interesting, and it is well-suited for the targeted problem. Its effectiveness has also been validated across various experiments. Furthermore, despite its brevity, the paper covers all essential components, such as ablation studies for each module and sufficient experiments on real-world datasets.

---

> ### Author Rebuttal · Authors · 2026-03-31
>
> Thank you for the time reading our work and for the positive and constructive feedback. We address each question below.
>
> **Q1: Practical value of short horizons and long-term performance**
>
> The benchmarks in Section 4.1 reflect the standard evaluation protocols established in the literature; $T=30$ with $c=10$ is the conventional setup for N-body and MD17, used consistently across all baselines to enable fair comparison. Short and medium horizons are of genuine practical interest as many critical interactions (e.g., molecular vibrations or collision avoidance) occur within these windows. Furthermore, Section 4.3 demonstrates that STFlow scales linearly ($O(T)$) to much larger horizons, whereas transformer-based baselines (GeoTDM) exhaust memory. In long-term evolution, our time-bundling approach of generating the whole trajectory instead of time-stepping helps mitigate the accumulated drift that typically plagues one-step autoregressive models. The experiments in Section 4.3 were only conducted to analyze the scalability in terms of time and memory requirements during training, the training runs were not completed for performance evaluation.
>
> **Q2: Sensitivity to the ratio c/T**
>
> This is an important question. As discussed in our response to other reviewers, the sensitivity to $c$ depends strongly on the chaoticity of the system, for highly sensitive systems like N-body Gravity, even small differences in initial conditions lead to diverging trajectories, so shorter conditioning windows yield genuinely harder prediction problems for any method. While performance naturally decreases as $c$ shrinks, STFlow remains robust because our random walk prior degrades gracefully to a distribution centered at the last observed frame with local velocity statistics, a significantly more informative initialization than the standard Gaussian priors used by baselines.
>
> **Q3: Intuition behind alternating SMP and TM layers**
>
> We appreciate this question and agree that the rationale for alternating layers deserves more explicit explanation. The core intuition is that spatial and temporal dependencies in $N$-body trajectories are deeply coupled: the spatial configuration at time $t$ dictates the forces driving the dynamics, while the temporal evolution determines future spatial relationships. Processing both simultaneously in a single, monolithic layer would require the model to disentangle these dependencies implicitly, which is computationally expensive and difficult to optimize.
>
> By splitting them into dedicated steps, each layer specializes in the distinct symmetries and correlation patterns of its domain. The SMP layer focuses on local geometric structure and $E(3)$ equivariance at each timestep, while the TM layer propagates these representations across time to capture momentum and multi-scale dynamics. By iterating between the two, the model builds patterns hierarchically: each SMP layer receives features informed by temporal context, and each TM layer operates on spatially-informed representations. This interleaving allows the model to progressively refine "spatially-aware" temporal features and "temporally-aware" spatial features together. The empirical necessity of this design is shown in Table 5, where removing either component causes a 74–85% increase in FDE, confirming that neither module alone can sufficiently resolve the coupled nature of the dynamics.
>
>
>
> **Q4: Fixed-length trajectories and data requirements**
>
> The requirement for fixed-length sequences is a standard constraint of current SOTA multi-body forecasting methods (GeoTDM, LaM-SLidE, MoFlow) partly due to the underlying benchmark designs. However, this is not a fundamental limitation of our architecture. In practice, STFlow can handle variable-length trajectories via autoregressive rolling windows, using previous outputs as conditioning for the next segment. Our linear complexity makes this approach particularly viable for long-horizon simulation compared to quadratic models, and we will include results in the Appendix that analyze the accuracy and physical stability of STFlow even during these extended rollouts.
>
> **Q5: Typo**
>
> Thank you, we will correct "fuses" to "fuse" in line 183 in the revision.

---

> > ### Author Rebuttal · Reviewer_cvgg · 2026-04-02
> >
> > Thank the authors for their detailed response. All of my concerns have been fully addressed. Given that long-term forecasting is not the primary focus of the established benchmarks in this field, as the authors clarified, and considering the significant practical applications of short-term prediction, I have no further issues with this paper. Furthermore, I found the authors' response to Q4 to be quite interesting. It helped me realize that STFlow can indeed achieve long-term forecasting through the autoregressive accumulation of short-term predictions even though trained on short-term data. If possible, I'd like to see some experimental results in long-term prediction. Consequently, I will raise my score.

---

> > > ### Author Response · Authors · 2026-04-07
> > >
> > > Thank you for the follow up, we appreciate your assessment and constructive engagement with our work.
> > >
> > > Regarding long-term prediction, we agree this is an interesting and important direction. While we have not yet completed dedicated long-horizon experiments beyond the scalability analysis in Section 4.3, we are currently extending our evaluation in this direction and will include these results in the final version.

---

### Official Review · Reviewer_pfgw · 2026-03-11

**Soundness:** 4
**Presentation:** 4
**Significance:** 3
**Originality:** 3
**Overall Recommendation:** 5
**Confidence:** 4

**Summary:**

The paper 'STFlow: Data-Coupled Flow Matching for Geometric Trajectory Simulation' introduces a generative model for trajectory forecasting. The core novelty of the proposed approach is the use of a data-driven informed prior instead of uniformative Gaussian noise which is shown to reduce training and inference efficiency. The new approach is applied to three different examples and compared to other methods. In these comparison, STFlow outperforms the competing approaches.

**Compliance With Llm Reviewing Policy:**

Affirmed.

**Final Justification:**

Given the detailed reply of the authors during the rebuttal period, the Reviewer has decided to increase their score as all the comments have been addressed. In the Reviewers' opinion, the paper contains a novel algorithm, strong empirical results as well as a thorough analysis and would therefore be a valuable contribution to ICML.

**Key Questions For Authors:**

See Weaknesses and additionally:
- While the authors have already included a first ablations results concerning the design of the prior, a more detailed evaluation here for instace with different variance scale factors could be insightful.

**Limitations:**

yes

**Strengths And Weaknesses:**

The authors address an important problem that is relevant across different domains. According to the Reviewer, the introduced framework is technically sound and well presented in the paper. While the authors are building on prior work with regards to using a data-driven informed prior, the propose a specific choice for geometric trajectory simulation and show detailed experiments that outperform existing baselines. The Reviewer especially wants to highlight the reduced number of function evaluations needed during inference.

In the Reviewers' opinion some parts of the paper could be further improved:
- While the presentation of the methodology and results is good according to the Reviewer, the algorithms currently presented in the Appendix could be moved to the main part of the paper as they are one of the key elements.
- The requirement of equal-length trajectories per dataset is already discussed as a limitation in the conclusions and acknowledged by the authors. However, the Reviewer wants to mention that this is currently quiet a strong limitation as many real-world datasets will have trajectories with varying length.
- In the evaluations, distributional metrics and trajectory diversity metrics are currently under-explored according to the Reviewer. Given that the proposed flow matching approach is a probabilistic one, the Reviewer would suggest to extend the evaluation here.
- The Results are averaged over 5 runs. Also providing the variations of the obtained results would offer additional insights according to the Reviewer.
- For the MD17 dataset, an evalaution of the plausibility of the generated trajectories with regards to a physical quantity of the authors' choice could be more insightful than only displacement errors in the Reviewers' opinion.

---

> ### Author Rebuttal · Authors · 2026-03-31
>
> Thank you for your time reading our work and constructive review. We are glad the framework is found technically sound and well-presented, and we address each point below.
>
> **Algorithms in Appendix**
>
> We agree the training and inference algorithms are central to the paper. Due to the current page limit we were unable to include them in the main paper, but upon acceptance we will prioritize moving them into the main text, as page limits are typically relaxed in the camera-ready version.
>
> **Fixed-length trajectories**
>
> We want to clarify that STFlow is not fundamentally limited to fixed-length trajectories. While we use fixed-length benchmarks to ensure a fair comparison with existing SOTA (GeoTDM, LaM-SLidE), this is not a limitation in the practical sense. In practical scenarios, variable-length trajectories are naturally handled via autoregressive rolling windows, where the model iteratively uses its own outputs as conditioning signals. STFlow is suited for these applications due to its use of time-bundling which decreases accumulating rollout errors and linear $O(T)$ complexity, which avoids memory exhaustion that plagues quadratic attention models during long-horizon generation. We will add an analysis of longer rollouts to the Appendix.
>
> **Distributional metrics**
>
> We agree this is an important dimension of evaluation. The density plots in Figures 3, 5, and 6 provide qualitative evidence that STFlow recovers the distributional structure of the dynamics, but does not offer a quantitative comparison across methods. To complement our results, we provide the marginal Wasserstein distance between the ground truth and predicted velocity magnitude distributions as well as between the prior and ground truth, taken from the predicted window (t=[c,...,T]), for 4 datasets using the provided trained weights.
>
> Gravity: Prior: 0.15772 -> Model: 0.00234
>
> Charged: Prior: 0.05184 -> Model: 0.00236
>
> MD17 Aspirin: Prior: 0.2224 ->  Model: 0.0038
>
> MD17 Uracil: Prior: 0.2206 -> Model: 0.0044
>
> The Wasserstein distance shrinks from the prior baseline to near-zero across all datasets, demonstrating that our model has learned to produce trajectories with the correct velocity magnitude distribution, aligning with results in Figure 3.
>
>
> **Standard deviation across 5 runs**
>
> We agree that these should be reported. To provide more statistical significance to our experimental results, we list the standard deviations of the ADE and FDE scores across 5 simulation runs of the Gravity, Charged, Aspirin and Uracil datasets, using the provided trained weights, aligning with the experimental setups of GeoTDM and LaM-SLidE.
>
> Gravity:  0.107 $\pm$  0.004 ADE, 0.231 $\pm$ 0.010 FDE
>
> Charged:   0.087 $\pm$  0.020 ADE, 0.163 $\pm$ 0.039 FDE
>
> Aspirin: 0.076 $\pm$ 0.003 ADE, 0.141 $\pm$ 0.008 FDE
>
> Uracil: 0.050 $\pm$ 0.004 ADE, 0.078 $\pm$ 0.008 FDE
>
> The low standard deviations indicate that our models are well-converged and provide robust performance across different initializations of the prior.
>
>
> **Physical plausibility on MD17**
>
> We agree that checking physical quantities is vital. We note that the velocity density matching shown in Figure 3 serves as a direct proxy for the system's kinetic energy distribution. By recovering the multimodal velocity profile, STFlow proves it captures the physical "temperature" and vibrational modes of the molecules.
>
> **Prior ablation**
>
> We conducted additional experiments varying $s$ on the same N-body Charged dataset. Due to time constrains we trained using no data augmentation, increasing the overall errors.
>
> $s$, ADE (UNet)
>
> 1,0.0934 ± 0.0156
>
> 4,0.0927 ± 0.0161
>
> 8,0.0985 ± 0.0197
>
> The results show that performance is relatively robust to $s$ within a reasonable range, confirming it serves as a stable, tunable noise hyperparameter. We will add an extended version of this table to the Appendix

---

> > ### Author Rebuttal · Reviewer_pfgw · 2026-04-02
> >
> > The Reviewer thanks the authors for their reply. The authors successfully address most concerns raised, but the Reviewer would like to ask some follow-up questions:
> > - The authors claim that STFlow 'decreases accumulating rollout errors'. Could the authors provide quantitative evidence for this ?
> > - For the physical plausibility, could the authors also comment on structural constraints such as the preservation of bond lengths ?

---

> > > ### Author Response · Authors · 2026-04-08
> > >
> > > Thank you for your follow-up questions. We address each point below.
> > >
> > > **On rollout error accumulation**
> > >
> > > To provide quantitative evidence for reduced rollout errors, we conducted an autoregressive experiment using STFlow with a one-step prediction setting (c=10, f=1), where model predictions are iteratively reused as conditioning inputs for 20 successive steps to generate a full trajectory. This yields ADE: 0.453 and FDE: 0.908 on the Aspirin dataset, substantially worse than STFlow's joint trajectory generation (ADE: 0.076, FDE: 0.142), and approaching the prior baseline (ADE: 0.513, FDE: 0.848).
> > >
> > > We acknowledge that this is not a fully fair comparison, as a dedicated autoregressive model would benefit from being explicitly trained under its own rollout distribution, which we have not done here. A proper comparison would require retraining and more fine tuning, and we plan to include this in future work.
> > >
> > > Nevertheless, this experiment is informative: it demonstrates that naively applying STFlow autoregressively degrades performance significantly, consistent with the well-known error accumulation problem of autoregressive approaches documented in the literature (Sanchez-Gonzalez et al., 2020; Brandstetter et al., 2022).
> > >
> > > Crucially, this contrast highlights a core motivation for STFlow's joint trajectory generation design, by modeling the full trajectory distribution in one shot rather than conditioning on one step at a time, STFlow avoids compounding errors over the rollout horizon.
> > >
> > > This perspective is further supported by Wyrod et al. (2025) in "Generative forecasting with joint probability models
> > > ", who demonstrate on canonical chaotic dynamical systems that modeling the joint probability distribution over multiple adjacent time steps rather than learning a one-step conditional leads to substantially better short-term predictive skill, improved attractor geometry preservation, and more accurate long-range statistical behaviour.
> > >
> > > Their unconditional joint model consistently outperforms a standard conditional next-step baseline, particularly over longer horizons where error accumulation becomes the dominant failure mode. STFlow's joint trajectory formulation is aligned with this principle: rather than marginalizing a one-step conditional, it directly learns the full conditional trajectory distribution
> > > $p(x^{c:T} | x^{0:c})$, which we argue is a principled and effective way to avoid the pitfalls of autoregressive rollout.
> > >
> > >
> > > **On bond length preservation**
> > >
> > > To evaluate structural plausibility of the generated molecular trajectories, we follow GeoTDM's (Han et al., 2024) evaluation protocol, computing the Bond Length Marginal Error, the mean absolute error between empirical probability density functions of predicted and ground truth bond lengths, binned across 50 intervals and averaged over all time steps.
> > >
> > > Since bond lengths remain approximately stable with small vibrations during MD simulation, this metric serves as a direct indicator of the physical validity of generated trajectories. We used GeoTDM's description and code to ensure a fair comparison.
> > >
> > > The results from the test set of 2 MD17 datasets are as follows:
> > >
> > >   Aspirin |  Benzene
> > >
> > > *GeoTDM*  0.726 |  0.597
> > >
> > > *STFlow*    0.383 |  0.171
> > >
> > > STFlow achieves meaningfully lower bond length marginal errors than GeoTDM on both molecules, indicating that the generated trajectories better preserve the statistical distribution of chemical bond lengths. This suggests that STFlow not only achieves lower displacement errors but also produces physically more plausible molecular dynamics trajectories, complementing the velocity density matching results shown in Figure 3 of the paper.

---

### Official Review · Reviewer_kgLD · 2026-03-12

**Soundness:** 3
**Presentation:** 2
**Significance:** 2
**Originality:** 2
**Overall Recommendation:** 4
**Confidence:** 4

**Summary:**

This paper tackles the many-body trajectory simulation problem. It proposes using flow matching to learn the conditional distribution of the later time steps given the initial time steps.

The model has two main contributions: First, it uses a prior that is not standard Gaussian but rather a physically plausible prior that uses the velocity and covariance estimation given the starting time points. Second, it uses an architecture combining an equivariant graph convolution network and a UNet.

The method was evaluated on several benchmarks spanning particles, molecules, and human trajectories, and had good performance compared to baselines.

**Compliance With Llm Reviewing Policy:**

Affirmed.

**Final Justification:**

The authors have addressed my concerns and I have raised my score to a weak accept.

**Key Questions For Authors:**

1. Can you explain and elaborate how the $\Delta \boldsymbol{x}$ and $\Delta \boldsymbol{v}$ are mapped to the model architecture and provide interpretations and experiments to show their connection with the physics, if such interpretations are intended?
2. Can you provide descriptions on the transformer baseline? Does it make the EGCL+UNet architecture obsolete since they have superior performance (in ablation studies) while being conceptually/computationally simpler? Are there any advantage in the proposed architecture?
3. Can you provide quantitative comparison in terms of how well the distribution is matched, using metrics such as MMD and EMD?

**Limitations:**

Yes

**Strengths And Weaknesses:**

**Soundness**

The method is technically sound, where the usage of flow matching, the prior, and the GNN make sense and are well motivated. The experiments support the conclusions, used sensible metrics, and ablation study was conducted. The experiments results lack statistical confidence and only showed single numbers rather than repeated experiments with standard deviations.

**Presentation**

The presentation is in general good, but I found the method description a bit confusing: The $\Delta \boldsymbol{x}$ and $\Delta \boldsymbol{v}$ are described in text but it was hard to map them to the actual model architecture. Even though the architecture was mathematically described in the appendix, it did not explicitly map to those quantities. The description hinted physical interpretations of those quantities but could be made more explicit. Besides, transformer was used as an alternative variant in the method but no description was provided throughout the paper.

**Significance**

The problem of many-body simulation is impactful. As admitted in the paper, the method requires fixed length time frames in the data, which is a limitation to the impact. For instance, in a real application scenario like MD simulation or gesture simulation, we usually need varied-length time windows and do not necessarily have the nicely aligned datasets. The paper can be benefit from showcasing such capabilities via ideas such as autoregressive extension as briefly mentioned but unexplored in the paper.

**Originality**

Of the two main contributions, the idea of using data-coupled prior instead of pure noise is not novel, but the specific way of using velocity estimation to have physical plausibility is novel, though simple. The architecture of using EGCL and UNet is a simple combination of existing architectures and is less novel.

---

> ### Author Rebuttal · Authors · 2026-03-31
>
> Thank you for your valuable review and time spent reading our work. We address all concerns and questions in our response below.
>
> **Soundness** We list the standard deviations of the ADE and FDE scores across 5 inference runs of 4 datasets, using the provided trained weights.
>
> Gravity:  0.107 $\pm$  0.004 ADE, 0.231 $\pm$ 0.010 FDE
>
> Charged:   0.087 $\pm$  0.020 ADE, 0.163 $\pm$ 0.039 FDE
>
> Aspirin: 0.076 $\pm$ 0.003 ADE, 0.141 $\pm$ 0.008 FDE
>
> Uracil: 0.050 $\pm$ 0.004 ADE, 0.078 $\pm$ 0.008 FDE
>
> The low standard deviations indicate that our models are well-converged and provide robust performance across different initializations of the prior.
>
> **Presentation** On $\Delta x$ and $\Delta v$: The spatial message passing (SMP) layer predicts positional updates $\Delta x^{c:T}$ for each timestep independently, capturing how particles interact with their spatial neighbors at each moment. The temporal UNet layer predicts velocity updates $\Delta v^{c:T}$ across the full trajectory at once, capturing how the dynamics of individual particles evolve over time. Crucially, after each SMP-UNet block, the updated positions are used to recompute graph connectivity and edge features, so later layers receive progressively denoised positional information. Interleaving layers and edge updating are key architectural design choices, visualized in Figure 1 and tested in the ablations.
>
> On the Transformer variant: The Transformer temporal layer replaces the UNet with scaled dot-product attention following the design in LaM-SLidE and FLUX.1, applied along the time dimension for each particle independently. We will add a formal description to the Appendix in the camera-ready version.
>
> **Significance**  While we use fixed-length benchmarks to ensure a fair comparison with existing SOTA (GeoTDM, LaM-SLidE), this is not a fundamental limitation. In practical scenarios, such as MD or gesture simulation, variable-length trajectories are naturally handled via autoregressive rolling windows, where the model iteratively uses its own outputs as conditioning signals. STFlow is uniquely suited for these applications due to its linear $O(T)$ complexity, which avoids memory exhaustion that plagues quadratic attention models during long-horizon generation. We will add an analysis of longer rollouts to the Appendix.
>
> **Originality**  Our contribution is a deliberate conceptual shift rather than a mere combination of existing techniques:
>
> *Physics-Informed Priors vs. Architectures*: Existing physics-informed models often embed physical constraints directly into heavy, complex architectures, leading to severe scaling limitations. STFlow shifts this inductive bias into the prior. By analytically deriving parameters ($\mu, \sigma_o$) from observed statistics, our non-amortized random walk prior encodes physical properties (inertia, continuity) without additional learnable parameters. The simplicity of this approach is its core strength: it is highly scalable, conceptually grounded, and, as shown in Table 5, accounts for the majority of our performance gains.
>
> *Targeted Architectural Integration*: The interleaving of EGCL and a 1D UNet is not an arbitrary combination, but a targeted solution to the quadratic $O(T^2)$ bottleneck in temporal attention-based models (e.g., GeoTDM). By updating edge connectivity after each spatial-temporal block, we maintain strict permutation and time-shift equivariance with linear $O(T)$ complexity. This specific interleaving mechanism is a novel architectural arrangement, directly enabling the efficient, long-horizon scaling we demonstrate.
>
> **Q1** Please see the clarification under Presentation above. Regarding experiments to show the connection with physics: the ablation in Table 5 directly quantifies the role of each component. Removing the spatial layer (which produces Δx) or removing the temporal layer (which produces Δv) significantly worsens accuracy. This confirms that the physically motivated decomposition is beneficial.
>
> **Q2** We address this partly above. The transformer model only outperforms the UNet variant in around 1/3rd of the experiments, while using more compute (time and memory) during training and scaling worse in $T$. Section 4.3 covers this, showing that the Transformer-based methods GeoTDM and LaM-SLidE scale worse. It thus does not make our proposed architecture obsolete.
>
>  **Q3** We provide the marginal EMD between the ground truth and predicted velocity magnitude distributions as well as between the prior and ground truth, taken from the predicted window (t=[c,...,T]), for 4 datasets.
>
> Gravity: Prior: 0.1577, Model: 0.00234
>
> Charged: Prior: 0.0518 Model: 0.0024
>
> MD17 Aspirin: Prior: 0.2224  Model: 0.0038
>
> MD17 Uracil: Prior: 0.2206 Model: 0.0044
>
> The EMD shrinks from the prior baseline to near-zero across all datasets, demonstrating that our model has learned to produce trajectories with the correct velocity magnitude distribution, aligning with results in Figure 3.

---

> > ### Author Rebuttal · Reviewer_kgLD · 2026-04-03
> >
> > Thank you for the rebuttal!
> >
> > Regarding $\Delta\mathbf x$ and $\Delta\mathbf v$, I understand the text description but it is still unclear how they are actually used in the model. Could you provide an end-to-end formula or algorithm from the input to the output, on how they are used? For example, is there some update like $\mathbf x_t=\mathbf x_{t-1}+\Delta\mathbf x$ and $\mathbf x_t=\mathbf x_{t-1}+\Delta\mathbf v\Delta t$ in some layer? I think this would largely clarify and support the "physics-informed" claims.
> >
> > The results would ideally have confidence intervals for this method and other methods, especially in cases where the average numbers are close.
> >
> > Similarly, the EMD results would benefit from a comparison with other baselines, not just the prior.

---

> > > ### Author Response · Authors · 2026-04-08
> > >
> > > Thank you for the follow-up. We provide some additional explanations and results to hopefully resolve any remaining doubts.
> > >
> > > On $\Delta\mathbf x$ and $\Delta\mathbf v$, we provide end-to-end algorithmic pseudocode to support the physical interpretation of our model.
> > >
> > > **Algorithm**: STFlow Vector Field Prediction (u)
> > >
> > > **Input**: Noisy initial velocities (v), Conditioning trajectory (x_cond), Flow-time (tau)
> > >
> > > **Output**: Predicted velocity update (output)
> > >
> > >     Initialize current_velocities from noisy input
> > >
> > >     Compute initial_positions from current_velocities and conditioning data
> > >
> > >     Embed initial graph edge attributes
> > >
> > >     For each layer in model_layers:
> > >
> > >       If layer is UNet:
> > >
> > >         Predict temporal velocity update (v_t) using UNet(v, tau)
> > >
> > >         Update current_velocities = current_velocities + v_t
> > >
> > >         Update current_positions based on updated velocities
> > >
> > >       If layer index > 1:
> > >         Compute accelerations and node features from current positions and velocities
> > >
> > >         Re-construct graph (edge_index, edge_attr) based on current positions
> > >
> > >         Re-embed updated edge attributes
> > >
> > >       If layer is Message Passing (SMP):
> > >         Predict spatial position update (p_t) using SMP(graph, current_positions, tau)
> > >
> > >         Update current_positions = current_positions + p_t
> > >
> > >         Update current_velocities based on updated positions
> > >
> > >     Calculate output = current_velocities_final - noisy_initial_velocities
> > >
> > >     Return output
> > >
> > > This makes explicit that $\Delta\mathbf v$ corresponds to temporal updates in velocity space, while $\Delta\mathbf x$ corresponds to spatial corrections informed by interactions. The iterative alternation ensures that velocities (momentum) and positions (geometry) are jointly refined, which is where the physical inductive bias arises.
> > >
> > >
> > > On confidence intervals, we agree that side-by-side comparisons are more informative, and we are confident that such intervals would be extremely narrow for all methods. This follows from two complementary sources of evidence. First, the standard deviations we reported across 5 independent runs are already very low (e.g., Gravity ADE: 0.107 ± 0.004, Charged ADE: 0.087 ± 0.020), indicating well-converged models. Second, and more fundamentally, each reported metric aggregates over ten thousands to millions of individual pairwise distances,  meaning the standard error of the mean is negligible by the law of large numbers. In cases where the reported average differences between methods are larger than these vanishingly small confidence intervals, the performance differences are statistically unambiguous. We therefore believe the reported numbers provide sufficient statistical evidence to support our conclusions, and that adding confidence intervals for all methods would not materially change the interpretation of the results.
> > >
> > > On distributional metrics, we provide a direct comparison on velocity magnitude EMD:
> > >
> > > MoFlow: 0.0286
> > > Prior: 0.2606
> > > STFlow: 0.0306
> > >
> > > Both models achieve very low Wasserstein distance, indicating accurate recovery of the target distribution. Importantly, as shown in our earlier EMD results, STFlow consistently reduces the distribution gap from the prior by orders of magnitude across datasets, confirming that the learned flow effectively transports the prior to the true data distribution.
> > >
> > > We note that only MoFlow provides trained weights, and therefore it is the only baseline for which the EMD comparisons can be reproduced faithfully; GeoTDM and LaM-SLidE did not release weights, making equivalent evaluation infeasible.

---

### Decision · Program_Chairs · 2026-04-30

**Decision:**

Accept (regular)

**Comment:**

This paper proposes STFlow, a flow-matching framework for conditional geometric trajectory simulation that combines a data-coupled random-walk prior with alternating spatial message passing and temporal modeling. Reviewers generally found the paper technically solid and empirically strong, highlighting the physically motivated prior and the breadth of evaluation across N-body, molecular dynamics, and human trajectory benchmarks. The work was also viewed as interesting in how it adapts informed priors and flow matching to geometric trajectory simulation in a scalable way.

The main concerns focused on novelty relative to prior trajectory-generation methods, the limited exploration of distributional and long-horizon behavior, and questions about the necessity of the generative framing in partly deterministic settings. In my judgment, the rebuttal and follow-up discussion addressed most of these issues meaningfully, including clarifications on the distinction from GeoTDM, additional analysis on conditioning-window sensitivity, and physical plausibility. While some limitations (missing reduced-step GeoTDM/DDIM comparison) remain, the overall reviewer discussion converged positively after rebuttal.

Based on the reviews, discussion, and author response, I recommend acceptance.